# Quantification of protein abundance and interaction defines a mechanism for operation of the circadian clock

Alex A Koch[1†], James S Bagnall[1†], Nicola J Smyllie[2], Nicola Begley[1], Antony D Adamson[1], Jennifer L Fribourgh[3], David G Spiller[1], Qing-Jun Meng[1], Carrie L Partch[3], Korbinian Strimmer[4], Thomas A House[4], Michael H Hastings[2], Andrew SI Loudon[1]*‡

[1]Faculty of Biology, Medicine and Health, University of Manchester, Manchester, United Kingdom; [2]MRC Laboratory of Molecular Biology, Cambridge, United Kingdom; [3]Department of Chemistry and Biochemistry, University of California, Santa Cruz, Santa Cruz, United States; [4]Department of Mathematics, University of Manchester, Manchester, United Kingdom

*For correspondence:
andrew.loudon@manchester.
ac.uk

†These authors contributed
equally to this work

Present address: ‡Faculty of
Biology, Medicine and Health,
University of Manchester,
Manchester, United Kingdom

**Competing interest:** The authors declare that no competing interests exist.

**Abstract** The mammalian circadian clock exerts control of daily gene expression through cycles of DNA binding. Here, we develop a quantitative model of how a finite pool of BMAL1 protein can regulate thousands of target sites over daily time scales. We used quantitative imaging to track dynamic changes in endogenous labelled proteins across peripheral tissues and the SCN. We determine the contribution of multiple rhythmic processes coordinating BMAL1 DNA binding, including cycling molecular abundance, binding affinities, and repression. We find nuclear BMAL1 concentration determines corresponding CLOCK through heterodimerisation and define a DNA residence time of this complex. Repression of CLOCK:BMAL1 is achieved through rhythmic changes to BMAL1:CRY1 association and high-affinity interactions between PER2:CRY1 which mediates CLOCK:BMAL1 displacement from DNA. Finally, stochastic modelling reveals a dual role for PER:CRY complexes in which increasing concentrations of PER2:CRY1 promotes removal of BMAL1:-CLOCK from genes consequently enhancing ability to move to new target sites.

## Editor's evaluation

The transcriptional negative feedback loop of the mammalian circadian clock is mainly regulated by interactions among BMAL1, CLOCK, PER1/2 and CRY1/2 in the nucleus. While the binding of CRY with BMAL1:CLOCK is known to block the transcriptional activity of BMAL1:CLOCK and the binding of PER:CRY dissociates BMAL1:CLOCK from DNA have been known, our understanding is limited in qualitative level. Koch et al., quantified the dynamic interactions among the core clock molecules such as their diffusion coefficients, binding affinity, and abundances in the nucleus. This greatly improves our understanding of the mammalian circadian clock. Importantly, this dynamic information is incorporated via a mathematical model to understand BMAL1-CLOCK binding to the target site (e.g., circadian proteins operate within an optimal range to modulate E-box binding), providing a coherent view on the mechanism driving the oscillation.

## Introduction

The 24 hr light-dark cycles inherent to our planet have led to the evolution of molecular circuits capable of conveying time of day information, commonly known as circadian clocks. In mammals,

cell-autonomous circadian clocks operate in virtually all cells across tissues and enables coordination of numerous biological processes, including metabolism, immunity, and cell cycle progression (*Bhadra et al., 2017*; *Gibbs et al., 2014*). Autonomous cellular clocks are characterised by transcription-translation feedback loops (TTFLs), leading to cycles in protein and mRNA tuned to the 24 hr rhythms of the day-night cycle. Central to the mammalian circadian clock is the heterodimeric transcription factor comprised of CLOCK (circadian locomotor output cycles protein kaput) and BMAL1 (brain and muscle ARNT-like 1) that searches the genome to bind consensus sequence E-box sites (CANNTG), inducing expression of several hundred clock-controlled output genes every day. Targets include key circadian negative feedback regulators, Period (Per1, Per2, Per3), Cryptochrome (Cry1 and Cry2) and a secondary loop regulated by Nr1d1 and Nr1d2 (*Buhr and Takahashi, 2013*; *Gekakis et al., 1998*; *Huang et al., 2012*; *Liu et al., 2008*). These proteins act to repress the activity of CLOCK:BMAL1 to form a delayed negative feedback loop driving daily oscillations. In a current model, it is proposed that PER and CRY proteins dimerise to form a repressive complex with CK1 (casein kinase 1) to promote the removal of CLOCK:BMAL1 from DNA and thereby repress transactivation of target genes, while CRY1 independently binds the PAS domain core of CLOCK:BMAL1 and the BMAL1 transactivation domain leaving DNA binding intact whilst repressing recruitment of additional transcriptional coactivators (*Chiou et al., 2016*; *Xu et al., 2015*). An additional feedback loop is conferred by the protein REV-ERBα, which operates as a transcriptional repressor of Bmal1, resulting in a cycle of BMAL1 protein abundance (*Liu et al., 2008*).

Ultimately, a prerequisite for generation and output of cellular circadian rhythms is the ability of a finite pool of CLOCK:BMAL1 heterodimer protein to bind rhythmically to specific target sequences leading to the regulation of circadian gene expression in cells (*Koike et al., 2012*). Currently, we have very little insight into the quantitative biology of this process. Heterodimeric formation of transactivating and repressive complexes is a well-defined feature of the circadian molecular circuit, including the formation of CLOCK:BMAL1 and PER:CRY complexes (*Chiou et al., 2016*; *Huang et al., 2012*; *Xu et al., 2015*). Recently, PER:CRY proteins have been described as part of very large macromolecular complexes within the cell (*Aryal et al., 2017*). We have previously visualised several core circadian proteins, and from this measured the spatiotemporal profile and protein abundance for BMAL1 and PER2 (*Smyllie et al., 2016*; *Yang et al., 2020*). PER2 was found to cycle with a maximum amplitude of 12,000 copies per cell in fibroblasts and without circadian gating of nuclear localisation, contrary to observations in the *Drosophila* clock (*Shafer et al., 2002*; *Smyllie et al., 2016*). Only a relatively small amount of CRY1 is needed to localise PER2 to the nucleus, as shown in live SCN studies, with PER2 localisation remaining predominantly nuclear throughout the day (*Smyllie et al., 2022*). A recent study using a cancer cell line model has also shown that CRY1 protein remains nuclear at all circadian phases and at markedly higher abundance than its partner protein PER2 (*Gabriel et al., 2021*). In order to gain insight into the operation of core circadian clock proteins, we generated a genetically modified mouse in which CRY1 has been C-terminally fused with a fluorescent protein. We crossed this line to a previously described strain of mice expressing fluorescent-tagged BMAL1. We then used advanced imaging in both ectopically transformed cell lines and endogenously modified mice to characterise governing parameters in the regulation of CLOCK:BMAL1 DNA binding, including repression by PER2 and CRY1. We constructed mathematical models of the complex interactions and phased timings from multiple molecular species and experimentally inaccessible complexes, demonstrating how DNA binding in the peripheral circadian clock is regulated.

Using a combination of mathematical modelling and experimental validation, our data reveal that high-affinity interactions between circadian protein complexes serve to offset the low abundances of circadian proteins. In this way, the abundance of key components of the molecular clockwork is positioned optimally to regulate E-box binding. This is partly facilitated through PER2:CRY1 mediated displacement of CLOCK:BMAL1, such that PER2 protein serves a dual role, acting as both a component of the negative feedback arm but also to redistribute CLOCK:BMAL1 to new target sites. Thus, the stochiometric balance of PER:CRY with CLOCK:BMAL1 is critical for the elucidation of the full cellular circadian repertoire.

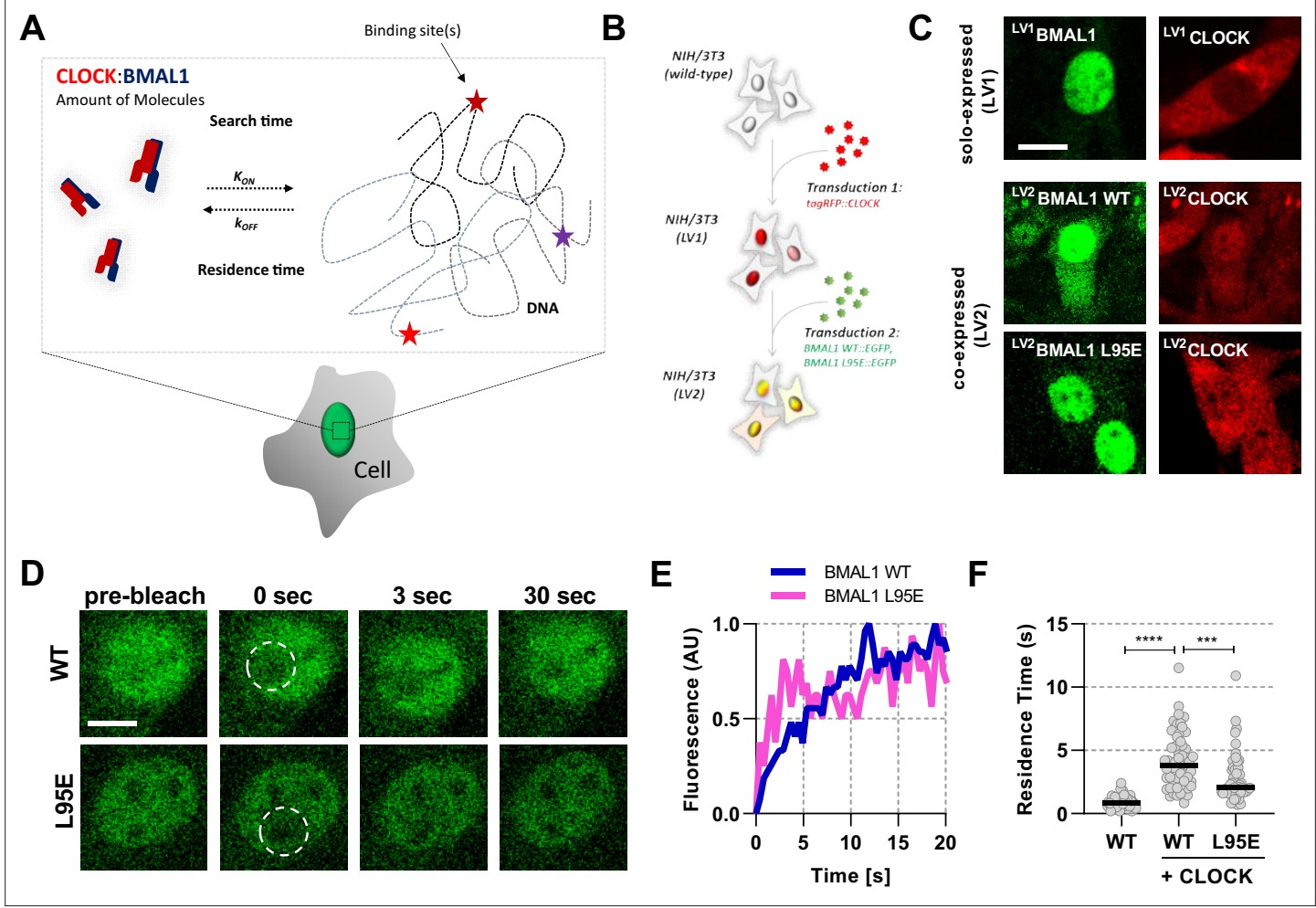

**Figure 1.** Short-lived DNA binding of BMAL1 and CLOCK. (**A**) Schematic representation of parameters regulating CLOCK:BMAL1 dimers binding to target DNA sites. (**B**) NIH/3T3 cells are either singularly or sequentially transduced to express fluorescent fusions with CLOCK or BMAL1 (wildtype and mutant variants).(**C**) Confocal microscopy images of cells solo-expressing (LV1) either tagRFP::CLOCK or BMAL1::EGFP or co-expressing (LV2) them together (including BMAL1 L95E DNA-binding mutant). (**D**) Confocal microscopy images for photobleaching of <sup>LV2</sup>BMAL1::EGFP-RFP::CLOCK labelled cells, either with wild-type or BMAL1 L95E DNA binding mutant. Images show nuclei and highlight region of bleaching. (**E**) Representative fluorescence recovery curves of bleach region for B. following normalisation. (**F**) Residence time calculated as the inverse of kOFF ($s^{-1}$), determined from fitting the recovery data with a single component binding model (n = 69, 58, and 51 cells). Bar represents median values. Source data for panel F available as *Figure 1—source data 1*.

The online version of this article includes the following source data and figure supplement(s) for figure 1:

**Source data 1.** BMAL1 residence times.

**Figure supplement 1.** Ectopically expressed mRNA is the major form in a lentivirus transduced system.

**Figure supplement 1—source data 1.** Summary statistics.

**Figure supplement 2.** Binding plays a significant role in BMAL1 mobility.

# Results

## BMAL1 determines nuclear localisation and mobility of CLOCK

To quantify the properties of BMAL1 and CLOCK proteins (*Figure 1A*), we first used NIH/3T3 fibroblasts expressing fluorescent fusion proteins via a ubiquitin ligase C promoter, delivered by lentiviral transduction either singly (LV1) or as two sequential transductions (LV2) (*Figure 1B*; *Bagnall et al., 2015*). Expression of the transgene was in >10 fold excess over the native unfused protein, as determined by single molecule Fluorescence In Situ Hybridisation (*Figure 1—figure supplement 1B*). Confocal microscopy of tagRFP::CLOCK or BMAL1::EGFP showed BMAL1 expression to be strongly

localised to the nucleus, whereas CLOCK was predominantly cytoplasmic when expressed alone (*Figure 1C*). Co-expression of both proteins in the same cells caused localisation of tagRFP::CLOCK to move to the nucleus, in agreement with earlier studies which showed cytoplasmic CLOCK localisation in BMAL1-deficient cells, and that circadian regulation of nuclear localisation of CLOCK correlated with BMAL1 availability (*Kondratov et al., 2003*; *Stratmann et al., 2012*). We also transduced cells with a fluorescent fusion of a DNA-binding mutant of BMAL1, in which a leucine is substituted for glutamic acid in the basic helix-loop-helix region of the protein; referred to as L95E. The mutant BMAL1 also re-localised tagRFP::CLOCK protein to the nucleus from the cytoplasm in an manner equivalent to WT BMAL1 (*Figure 1C*; *Huang et al., 2012*). We next performed Fluorescence Recovery After Photobleaching (FRAP) experiments to test the impact of CLOCK on the recovery dynamics of a bleached nuclear region of BMAL1::EGFP, by comparing responses with or without co-expressed tagRFP::CLOCK (*Figure 1D–F*). BMAL1 recovery half-life was found to be insensitive to the diameter of the bleached region, indicating that binding contributes to the recovery profile rather than this being a solely diffusion-led process (*Figure 1—figure supplement 2*; *Sprague and McNally, 2005*). Reaction binding equations were fitted to determine the rate of dissociation, $k_{OFF}$, for BMAL1::EGFP, the reciprocal of which equates to an average characteristic duration of binding or residence time. Residence time of BMAL1::EGFP was increased in the presence of tagRFP::CLOCK (p < 0.0001), consistent with a requirement for CLOCK to bind DNA (*Figure 1F*). The mean residence time for the fluorescent CLOCK:BMAL1 complex was 4.13 s (95% CI, 0.57), a value consistent with DNA residence times for similar transcription factors (*Hettich and Gebhardt, 2018*; *Stratmann et al., 2012*). Using the L95E DNA-binding mutant protein, we saw significantly reduced residence time of 2.83 s (95% CI, 0.54; p = 0.0002). Notably, the initial publication of the BMAL1 L95E mutant showed a twofold reduction in PER2::LUC expression and so suggests a strong relationship between DNA binding and transcriptional output (*Huang et al., 2012*).

To investigate this further we used Fluorescence (Cross) Correlation Spectroscopy (F(C)CS) (*Yu et al., 2021*), a technique used to determine live-cell concentration and diffusion properties of individually fluorescent-labeled BMAL1 and CLOCK proteins, as well as their interactions when co-expressed (*Figure 2A*). A normal diffusion model fitted the majority of data collected from cells expressing free EGFP or nuclear only NLS::EGFP proteins, as previously reported (*Dross et al., 2009*), whereas anomalous diffusion models – sub-diffusion caused by a range of factors such as DNA interactions and molecular crowding – accounted for a > 20% fraction, which in this instance may be explained by molecular crowding (*Tsekouras et al., 2015*). In comparison, for the fusion proteins, anomalous diffusion models accounted for >80% of all BMAL1 data sets (*Figure 2—figure supplement 1*). We used an anomalous diffusion model for all further analyses of circadian proteins to calculate diffusion coefficients and protein concentrations.

Singly expressed fluorescent BMAL1 and CLOCK were found to diffuse rapidly with a median coefficient of 9.2 $m^2 s^{-1}$ (SD, 3.3) and 12.6 $m^2 s^{-1}$ (SD, 6.1), respectively. In contrast, co-expression significantly reduced the rate of diffusion to 1.9 $m^2 s^{-1}$ (SD, 1.3; p < 0.0001) and 4.7 $m^2 s^{-1}$ (SD, 3.2; p < 0.0001) for BMAL1 and CLOCK ,respectively (*Figure 2B*). The L95E mutant diffused more rapidly than WT BMAL1, consistent with fewer interactions with DNA in the nucleus (*Figure 2C*). When co-expressed, WT BMAL1 and CLOCK exhibited a 2:1 concentration ratio in the nucleus (*Figure 2D*, *Figure 2—figure supplement 2E*), presumably arising from a combination of differences in protein turnover, shuttling and direct interaction. F(C)CS was then used to observe this interaction and determine a live-cell dissociation constant ($K_D$; reciprocal measure of affinity) (*Krieger et al., 2015*). A positive cross correlation curve was observed between BMAL1::EGFP and tagRFP::CLOCK that was not apparent in cells expressing NLS::EGFP with tagRFP::CLOCK (*Figure 2E*). To calculate $K_D$, we fitted a one-site saturating binding curve to the relationship between heterodimer and monomer which yielded a value of 148 nm (SD, 9.8) for WT BMAL1::EGFP and tagRFP::CLOCK (*Figure 2F*). The $K_D$ was measured for cells with the reverse fluorescent protein labelling, namely EGFP::CLOCK and BMAL1::-tagRFP, finding similar a value of 145 nm (SD, 4.8), although a stronger interaction was found in vitro by surface plasmon resonance (*Figure 2—figure supplement 2F*). Previous work found that the V435R mutation of BMAL1 in the PAS-B domain, leads to reduced dimerisation with CLOCK (*Huang et al., 2012*). We used the V435R mutation to confirm our F(C)CS measurements by co-expressing V435R-BMAL1 and WT-CLOCK. This elicited a ≈1.5-fold reduction in interaction affinity, resulting in a $K_D$ of 201 nm (SD, 14) (*Figure 2G*) and a reduction from 2:1 to a 4:1 ratio of BMAL1 and CLOCK in

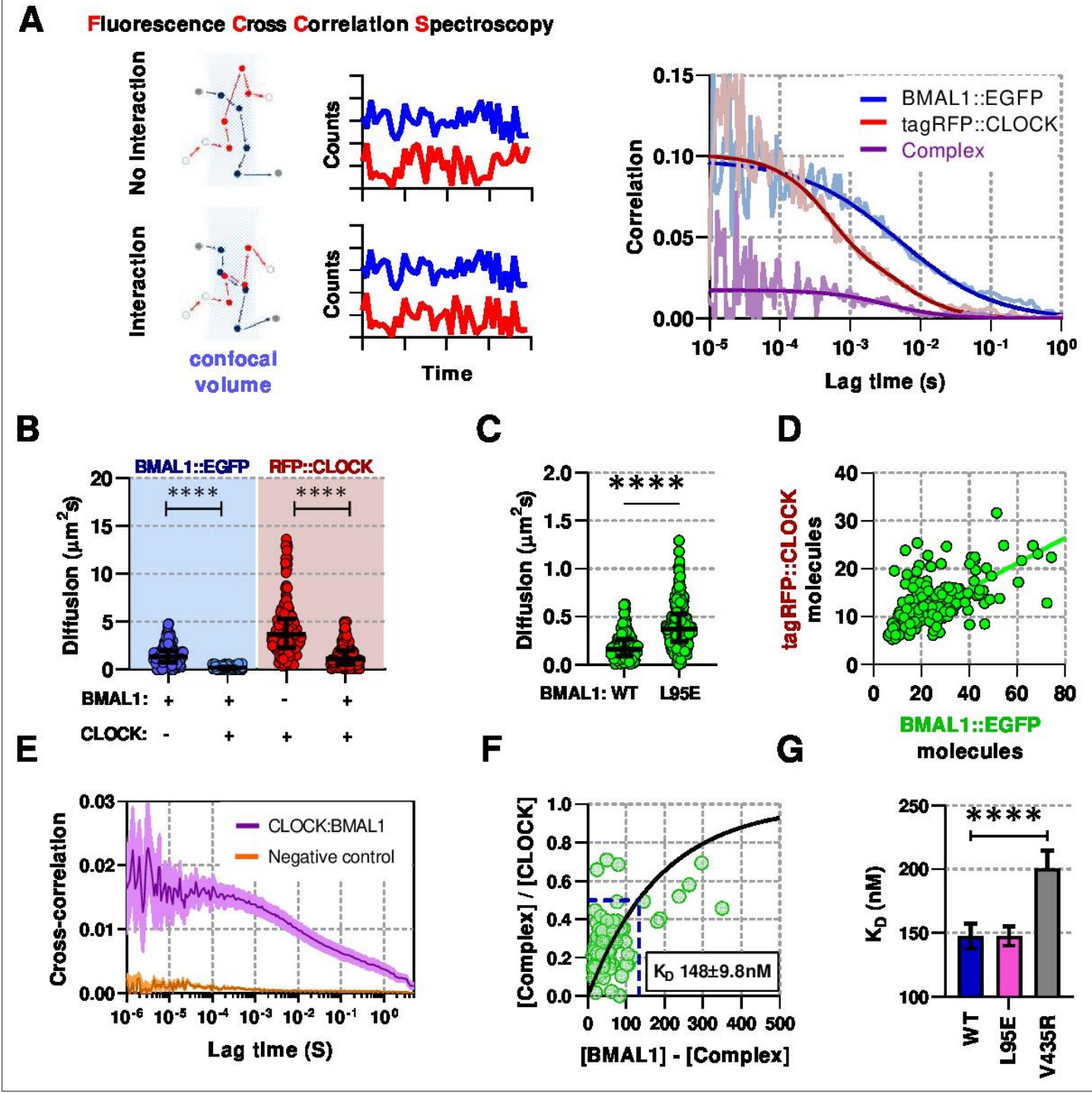

**Figure 2.** Live-cell interaction measurements demonstrate BMAL1 and CLOCK mobility is regulated by dimerisation and DNA binding. (**A**) Schematic of confocal volume used in FCCS with corresponding photon count traces. Interaction may be seen by correlation between both channels. Representative auto- and cross- correlation data showing raw data and fit lines for monomeric and complexed fluorescent proteins. (**B**) FCS data showing diffusion for BMAL1 and CLOCK in solo- and co-expressed conditions (n = 173, 152, 198, and 185 cells). (**C**) FCS results for BMAL1::EGFP diffusion for NIH/3T3 cells that co-express tagRFP::CLOCK. Data shown is for comparison of BMAL1 as either wild-type of L95E DNA-binding mutant. Bars show median and interquartile range. (**D**) Correlation of nuclear protein quantification showing relationship of BMAL1::EGFP with tagRFP::CLOCK for both wildtype and DNA binding mutant (n = 221 cells from three biological replicates). (**E**) Average cross-correlation curves for BMAL1::EGFP (WT) with tagRFP::CLOCK (n = 140) compared to a non-interacting control of NLS::EGFP co-expressed with tagRFP::CLOCK (n = 408). (**F**) Dissociation plot from FCCS data for BMAL1::WT and tagRFP::CLOCK. (**G**) Summary of calculated dissociation constants across all conditions, including BMAL1 dimerisation mutant, V435R

*Figure 2 continued on next page*

*Figure 2 continued*

(n = 156, 274, and 244). Mann-Whitney non-parametric test to determine significance (values are denoted as p > 0.05 ns, p < 0.05 *, p < 0.01 **, p < 0.001 *** and p < 0.0001 ****). Source data for panels B,C available as *Figure 2—source data 1* and panel E as *Figure 2—source data 2*.

The online version of this article includes the following source data and figure supplement(s) for figure 2:

**Source data 1.** BMAL1 and CLOCK FCS diffusion rates.

**Source data 2.** BMAL1 and CLOCK paired FCS concentrations.

**Figure supplement 1.** Anomalous diffusion best fits protein movement.

**Figure supplement 2.** Fluorescent BMAL1 and CLOCK proteins behave similarly when colours are swapped.

the nucleus (*Figure 2—figure supplement 2E*). In contrast, the BMAL1 L95E DNA binding mutant showed no difference in interaction affinity compared to WT BMAL1 protein. These data demonstrate that BMAL1 is a critical determinant of the localisation, mobility and concentration of CLOCK in the nucleus.

From this, we can infer the abundance of nuclear CLOCK from measurements of BMAL1, and make use of available endogenously labelled Venus::BMAL1 mice to measure remaining DNA binding parameters. First, we confirmed our cell line measurements for binding rates and diffusion using the Venus::BMAL1 mice (*Yang et al., 2020*), finding that they remain within a similar range across a number of primary cell types, including macrophages and pulmonary fibroblasts (*Figure 3—figure supplement 1*). We also measured protein number of the endogenous BMAL1, observing that total copies per nucleus vary from 1000 to 10,000 between individual cells, likely due to desynchrony and differing nuclear volumes. Moreover, a large overlap in nuclear copy numbers was observed across all cell types despite substantial changes in the mean. These data are critical in our understanding of the ratio of BMAL1 to target sites, effectively determining the capacity to regulate the full repertoire of target genes within a specific cell type.

## Quantification of strong rhythmic interaction of BMAL1 with CRY1

The ability to measure BMAL1 amounts to infer copy number of CLOCK, allows us to measure other critical pairings with BMAL1. This includes the repressive action of CRY1 binding to CLOCK:BMAL1, resulting in reduced transactivation. To explore the interaction between BMAL1 and CRY1, we generated a genetically modified mouse in which CRY1 has been C-terminally fused with the red fluorescent protein mRuby3 using CRISPR-mediated genomic editing to insert the coding sequence, replacing the endogenous CRY1 stop codon (*Figure 3—figure supplement 3A*; *Bajar et al., 2016*; *Bennett et al., 2021*). First, to test any potential impact on circadian pace-making, we measured CRY1::mRuby3 fluorescence in whole-field organotypic SCN slices (*Figure 3—figure supplement 3B*) which exhibited regular cycles in red fluorescence with a period of 23.9 hr (SD, 0.6) (*Figure 3—figure supplement 3C*; *Smyllie et al., 2016*). Additionally, wheel running measurements of these mice confirmed normal behavioural rhythmicity (*Figure 3—figure supplement 2*). We next crossed these mice to the Venus::BMAL1 mouse line (*Yang et al., 2020*), previously inter-crossed with a PER2::LUCIF-ERASE background (*Yoo et al., 2004*) to provide an independent circadian phase-reference marker (referred to as BMAL1xCRY1 labelled mouse). Using isolated lung fibroblasts from BMAL1xCRY1 mice we assessed bioluminescence in response to dexamethasone (DEX) synchronisation, and observed 23.3 hr cycles (SD, 0.6) which were sustained for >4 days (*Figure 3—figure supplement Figure 3—figure supplement 3D, E*). From this, we are confident that the fluorescent fusion proteins do not disrupt the normal operation of the circadian pacemaker.

Using the same synchronisation approach, we then measured BMAL1xCRY1 fluorescence in single cells every 4 hr from 24 to 48 hr post-DEX synchronisation, using F(C)CS (*Figure 3A-B*). Both fluorescent signals were localised to the nucleus. Venus::BMAL1 showed a consistent diffusion pattern over a circadian cycle, with a mean diffusion coefficient of 0.58 $m^2 s^{-1}$ (SD, 0.03), whereas CRY1 mobility exhibited circadian variance, with slow diffusion 28 hr post-DEX and elevated diffusion rates 12 hr later (*Figure 3C*). Interestingly, this change in mobility is consistent with a binding to a mass equivalent to the molecular weight of PERIOD2. Peak protein concentrations of BMAL1 and CRY1 had an approximate and appropriate phase-separation of 8 hr (*Fustin et al., 2009*). Auto-correlation analyses

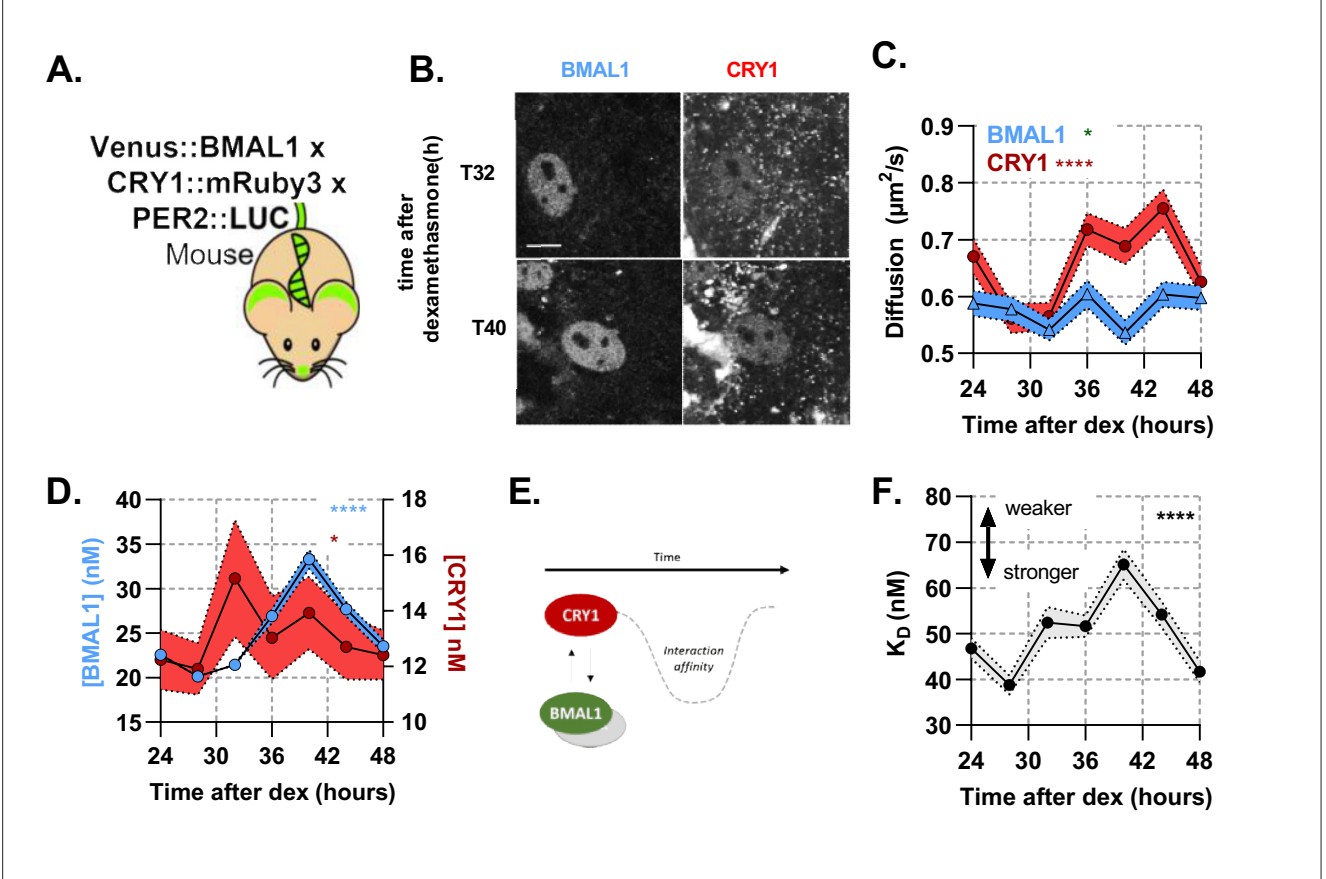

**Figure 3.** A rhythmic and strong interaction observed between slow-diffusing BMAL1 and CRY1 facilitates repression. (**A**) Schematic of triple-labelled mice from which primary lung fibroblasts were isolated (**B**) Confocal images of two cells shown for Venus::BMAL1 and CRY1::mRuby3 over time. FCS determined measurement for diffusion coefficient (**C**) and protein concentration (**D**) of Venus::BMAL1 and CRY1::mRuby3 (n = 136, 143, 173, 131, 158, 121, and 132; line shows the mean and error envelopes show the SEM). (**E–F**) Interaction strength between BMAL1 and CRY1 was also measured over time as illustrated by the schematic of affinity as well as plotted values of dissociation constant (error envelope shows the standard deviation). Kruskal-Wallis test used to determine significance (values are denoted as p > 0.05 ns, p < 0.05 *, p < 0.01 **, p < 0.001 *** and p < 0.0001 ****). Source data for panels B,C available as *Figure 3—source data 1*, *Figure 3—source data 2*, *Figure 3—source data 3*, *Figure 3—source data 4*.

The online version of this article includes the following source data and figure supplement(s) for figure 3:

**Source data 1.** CRY1 FCS diffusion rates.

**Source data 2.** BMAL1 FCS diffusion rates.

**Source data 3.** BMAL1 FCS concentration.

**Source data 4.** CRY1 FCS concentration.

**Figure supplement 1.** BMAL1 concentration and DNA binding parameters minimally vary across cell types.

**Figure supplement 1—source data 1.** CRY1::mRuby3 mouse genotyping.

**Figure supplement 2.** Generation of CRY1::mRuby3 mouse line.

**Figure supplement 3.** Triple endogenous labelled mice used to assay rhythms in SCN and peripheral lung fibroblasts.

revealed the concentration of BMAL1 is on average 1.92 fold (SD, 0.32) higher than CRY1, with a mean concentration of 29.3 nm and 13.4 nm respectively (equating to approximately 16,000 and 7000 molecules per nucleus), consistent with the range we reported earlier for PER2 (*Smyllie et al., 2016*). The amplitude of CRY1 was found to be shallow, cycling from 11.9 nm (SD, 5.7) at T28 to 15.2 nm (SD, 14.0) at T32, comparable to the approx. 25% amplitude observed for CRY1 in the SCN (*Figure 3— figure supplement 3C*). BMAL1 demonstrated a larger amplitude, cycling from 20.1 nm (SD, 7.1) at T28 to a peak of 33.3 nm (SD, 13.6) 40 hr after DEX (*Figure 3D*).

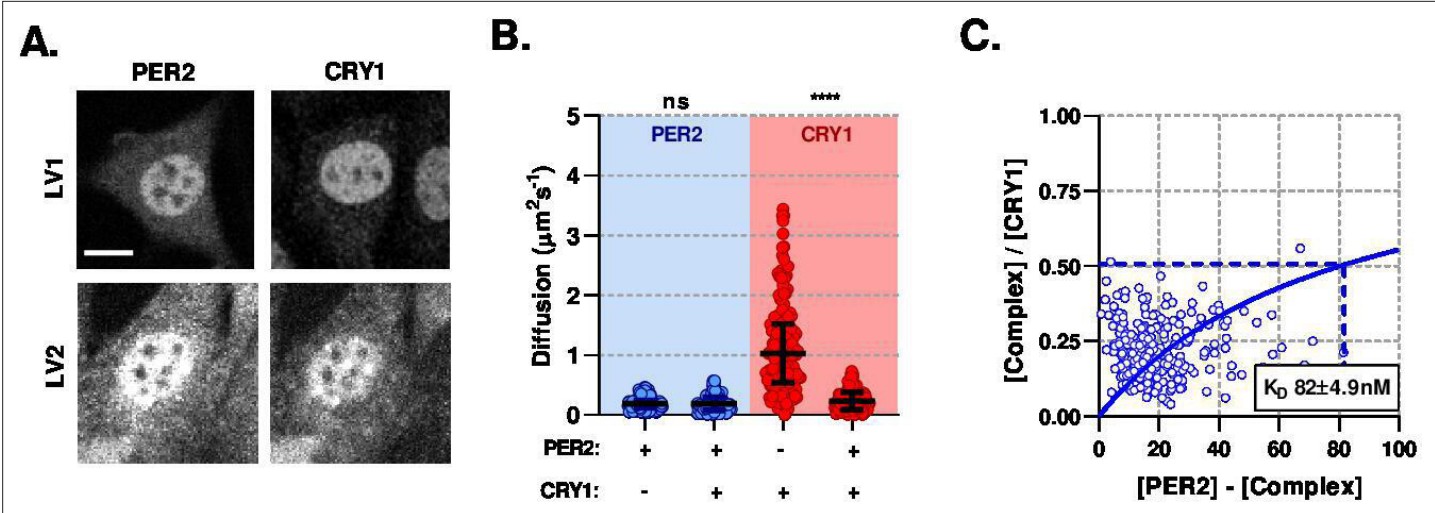

**Figure 4.** PER2 modulates CRY1 mobility via a high-affinity association. (**A**) Confocal images of transduced NIH/3T3 cells that either solo- or co- express PER2 and CRY1. (**B**) FCS data showing diffusion for PER2 and CRY1 in solo- and co-expressed conditions (n = 165, 174, 274, and 274 cells; diffusion rate means of 0.2, 0.2, 1.1, 0.2). (**C**) Dissociation plot from nuclear FCS measurements for EGFP::PER2 and CRY1::tagRFP (n = 274). Significance determined by Mann-Whitney test (values are denoted as p > 0.05 ns, p < 0.05 *, p < 0.01 **, p < 0.001 *** and p < 0.0001 ****). Source data for panels B available as *Figure 4—source data 1*.

The online version of this article includes the following source data and figure supplement(s) for figure 4:

**Source data 1.** PER2 and CRY1 FCS diffusion rates.

**Figure supplement 1.** CRY1 mobility is affected by co-expression with PER2.

We then analysed the interaction affinity between BMAL1 and CRY1 over time (*Figure 3F*). This interaction exhibited significant changes over a 24 hr cycle (p = < 0.0001), with the strongest interaction at T28, $K_D$ = 38.8 nm (SD, 2.1), and weakest at T40, $K_D$ = 65.1 nm (SD, 3.4) (*Figure 3F*). These profiles were found to correlate with the diffusion profile of CRY1 (*Figure 3C*). Intriguingly, the mean interaction strength between BMAL1 and CRY1 is >2 fold stronger than that between BMAL1 and CLOCK (*Figure 2G*). A similar relationship was found in vitro when measuring interactions using biolayer interferometry (*Fribourgh et al., 2020*). Although, these interaction measurements do not distinguish between whether either proteins are complexed with other partners, diffusion data is consistent with BMAL1 being bound to CLOCK, and are compatible with a model in which the low abundance of the CRY1 repressor is offset by a high affinity for the CLOCK:BMAL1 heterodimer.

The changes in the diffusion profile of CRY1 are consistent with its association with an additional binding partner, such as PER2, thereby altering the affinity of CRY1 for CLOCK:BMAL1 (*Fribourgh et al., 2020*; *Ye et al., 2014*). To measure the interaction between CRY1 and PER2 directly, we transduced NIH/3T3 cells with lentivirus so that cells constitutively express EGFP::PER2 or CRY1::tagRFP fusion proteins. In both cases, the expressed protein was found to localise predominately to the nucleus, although some cytoplasmic fluorescence was observed. When co-expressed, subcellular localisation was unchanged, although large punctate aggregates of signal were observed (*Figure 4A*). PER2 was found to have the slowest diffusion coefficient recorded within all our F(C)CS measurements, when in the non-aggregate space. PER2 mobility was not altered following co-expression with CRY1, whereas CRY1 exhibited reduced mobility in the presence of ectopic EGFP::PER2 (*Figure 4B*). The diffusion coefficient for CRY1 co-expressed with PER2 was similar to measurements of the endogenous protein (*Figure 3C*), suggesting PER2 and CRY1 exhibit similar stoichiometry within these cells. The anomalous diffusion model fit the majority of data sets, including ^LV1^CRY1, ^LV1^PER2, and ^LV2^PER2. However, normal diffusion models accounted for >50% of ^LV2^CRY1 correlation analyses suggesting a distinct change in CRY1 following interaction with PER2 (P < 0.0001), potentially from a loss of significant DNA binding of the CLOCK:BMAL1 complex (*Figure 4—figure supplement 1*). Best fit models for each data set demonstrated a strong affinity between PER2 and CRY1 with a $K_D$ of 81.8 nm (SD, 4.9) (*Figure 4C*) and consistent with previous in vitro measurements (*Schmalen et al., 2014*).

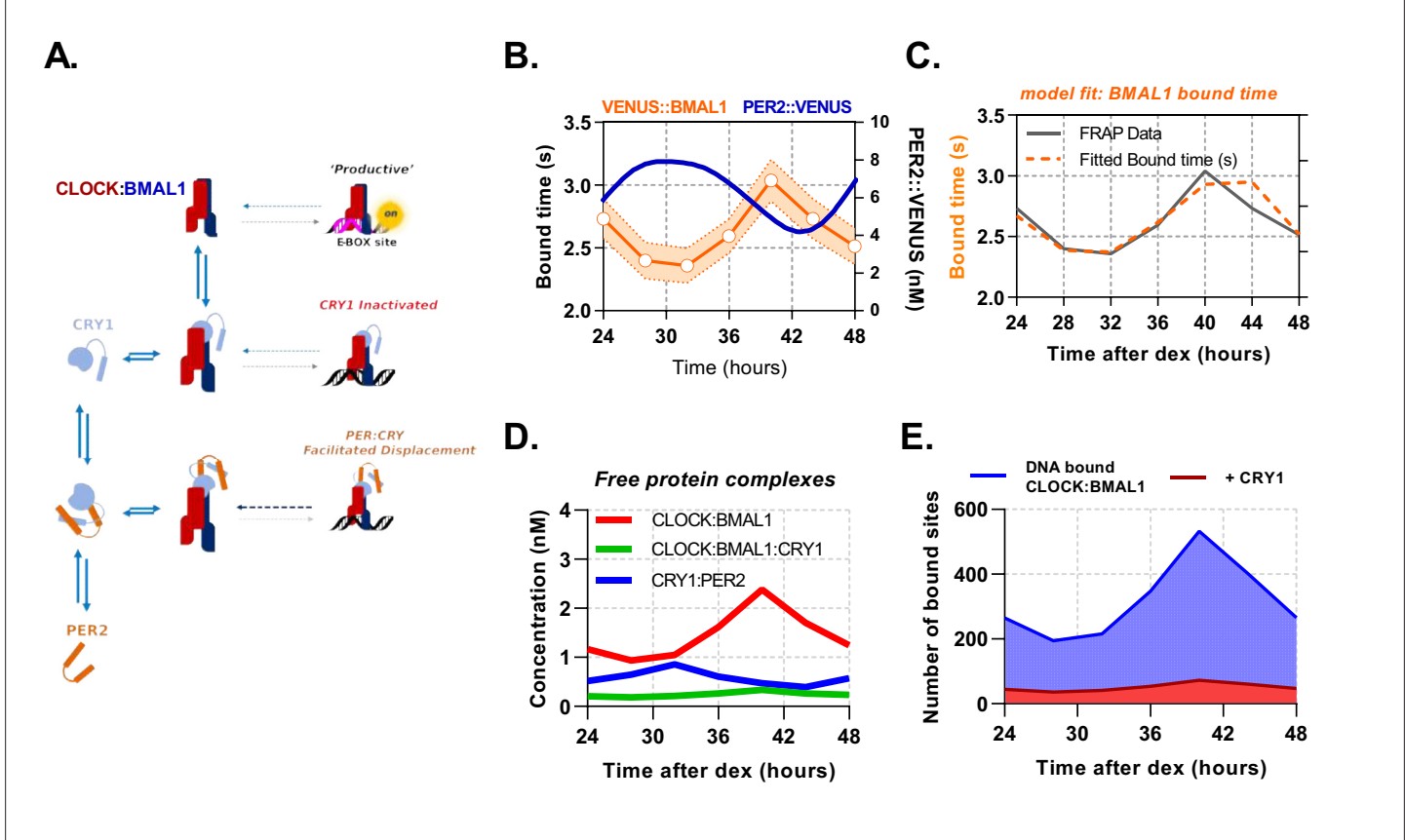

**Figure 5.** PER2 acts via CRY1 to mediate rhythmic displacement of CLOCK:BMAL1 from DNA. (**A**) Schematic representation of model topology used for the deterministic model of CLOCK:BMAL1 DNA binding. (**B**) Primary lung fibroblasts from BMAL1 x CRY1 x PER2 mice were synchronised with dexamethasone. Plot shows PER2 concentration as measured via FCS by *Smyllie et al., 2016* as well as mean BMAL1 binding time (showing SEM error envelope). Binding time was measured by confocal FRAP measurements performed on the Venus::BMAL1 fluorescence. Orange line shows the inverse of kOFF (s⁻¹), determined from fitting the recovery data with a single component model (n = 48, 70, 82, 63, 82, 64, and 65 cells). (**C**) ODE model was fit to FRAP binding data from E. and using a measured input for PER2 nuclear concentration previously determined in *Smyllie et al., 2016*. Model output showing (**D**) inferred nuclear concentrations for molecular complexes (**E**) and CLOCK:BMAL1 without and with CRY1 bound to target sites (see supplementary materials for parameters). Panel B has been adapted from Figure 3C from *Smyllie et al., 2016*.

The online version of this article includes the following figure supplement(s) for figure 5:

**Figure supplement 1.** ODE model of CLOCK:BMAL1 DNA binding using measured inputs and modelled perturbations.

Quantitative data are an enabling and often essential component of stringent mathematical modelling (*Bagnall et al., 2018*). Having quantified the necessary parameters, we then sought to use them in developing a mathematical model of CLOCK:BMAL1 DNA binding, with the aim of understanding how the multiple regulatory motifs of changing molecular concentrations, interactions and binding kinetics coalesce to regulate DNA binding and transcriptional activation of BMAL1. We explored multiple topologies that were able to fit BMAL1 binding rhythms, carrying the simplest model forward that incorporated our measured data. We modelled the system using a set of ordinary differential equations (ODEs) to depict a current understanding of the system; BMAL1 dimerises with CLOCK, which may subsequently bind and unbind to DNA target sites. To model repression, CRY1 may either inactivate CLOCK:BMAL1 via direct binding or, via dimerisation to PER2, form PER2:CRY1 (mimicking complexes with CK1) to displace CLOCK:BMAL1 from DNA (*Figure 5A*; *Chiou et al., 2016*; *Huang et al., 2012*; *Koike et al., 2012*; *Xu et al., 2015*). The latter would presumably lead to rhythmic changes in the residence time of BMAL1 and provide a sensible option to fit and complete the model.

To assess dynamic changes in inferred DNA binding rates of BMAL1, we isolated lung fibroblasts from BMAL1xCRY1 mice and determined the $k_{OFF}$ values by FRAP following DEX synchronisation.

**Table 1.** Summary of ordinary differential equation model parameters.
Model fit $\chi^2 = 7.46$.

**Input parameters**

| Parameter | Unit | Description | Value±SD |
|---|---|---|---|
| $K_D$(C:B - E-Box) | nM | CLOCK:BMAL1 - E-Box dissociation constant | 10 (*Huang et al., 2012*) |
| $K_D$(C-B) | nM | CLOCK - BMAL1 dissociation constant | $147.6 \pm 9.8$ |
| $K_D$(B-C1) | nM | BMAL1 - CRY1 dissociation constant | Time-point dependent, *Figure 3F* |
| $K_D$(C1-P2) | nM | CRY1 - PER2 dissociation constant | $81.8 \pm 4.9$ |

**Fitted parameters**

| Parameter | Unit | Description | Value±SD (Inverse Hessian eigenvalue of fit) |
|---|---|---|---|
| $k_{ON}$ | $nm^{-1}\,s^{-1}$ | CLOCK:BMAL1 - DNA binding on rate | $0.027 \pm 1.034\ (1.96)$ |
| $d_{ON}$ | $nm^{-1}\,s^{-1}$ | BMAL1 - CRY1 binding rate | $0.237 \pm 1.003\ (3.42 \times 10^{-2})$ |
| $a_{ON}$ | $nm^{-1}\,s^{-1}$ | PER2 - CRY1 binding rate | $6.34 \pm 1.00\ (6.79 \times 10^{-8})$ |
| $R_{OFF}$ | $s^{-1}$ | CLOCK:BMAL1:CRY1:PER2 - DNA unbinding rate | $(1.23 \pm 0.33) \times 10^1\ (1.00)$ |
| $b_{ON}$ | $nm^{-1}\,s^{-1}$ | CLOCK - BMAL1 binding rate | $9.17 \pm 1.29\ (1.00)$ |

**Derived parameters**

| Parameter | Unit | Description | Value±SD |
|---|---|---|---|
| $k_{OFF}$ | $s^{-1}$ | CLOCK:BMAL1 - DNA binding unbinding rate | $k_{ON} \times K_D$(C:B - E-Box) $= 0.27 \pm 10.34$ |
| $b_{OFF}$ | $s^{-1}$ | CLOCK - BMAL1 unbinding rate | $b_{ON} \times K_D$(C-B) $= (1.35 \pm 0.21) \times 10^3$ |
| $d_{OFF}$ | $nm^{-1}\,s^{-1}$ | BMAL1 - CRY1 unbinding rate | Time-point dependent, $d_{ON} \times K_D$(B:C1) |
| $a_{OFF}$ | $s^{-1}$ | PER2 - CRY1 unbinding rate | $a_{ON} \times K_D$(C1-P2) $= (5.19 \pm 0.88) \times 10^2$ |

Measurements of BMAL1 protein recovery were taken every 4 hr from 24 hr to 48 hr post-DEX, showing $k_{OFF}$ to be rhythmically regulated (*Figure 5B*). The BMAL1 $k_{OFF}$ profile was in antiphase to recordings of nuclear PER2 concentrations from *Smyllie et al., 2016* (*Figure 5B*). To fit all parameters to the model (*Table 1*), the measured concentrations of PER2 (*Smyllie et al., 2016*), CRY1 and BMAL1 were used as inputs, using data described in (Figs. *Figures 3D and 5C*). On/Off rates were constrained to measured dissociation constants from F(C)CS (*Table 1*), with the $K_D$ value between BMAL1 and E-box sites set at 10 nm, as measured by *Huang et al., 2012*. Using a mean value from multiple published ChIP-Seq data, the potential number of DNA target sites was set as 3,436 (*Table 2*).

The ODE model was then fitted by simulating FRAP so that a model-derived $k_{OFF}$ could be used against our experimental data (*Figure 5B*) via Chi$^2$ minimisation (Chi$^2$, 7.46) to mean and standard error on the mean (*Figure 5C*, *Table 1*). In order to confirm identifiability of the unknown parameters, we calculated the eigenvalues of the Hessian matrix of the fit, finding that it is non-singular and reasonably well conditioned (*Table 1*). Next, we used this model to infer

**Table 2.** BMAL1 ChIP reports.

| No. | Tissue | BMAL1 peaks | Reference |
|---|---|---|---|
| 1 | Liver | 2049 | *Rey et al., 2011* |
| 2 | Liver | 5952 | *Koike et al., 2012* |
| 3 | U2OS | 2001 | *Wu et al., 2017* |
| 4 | PECS | 2026 | *Oishi et al., 2017* |
| 5 | Liver | 4813 | *Beytebiere et al., 2019* |
| 6 | Kidney | 4034 | *Beytebiere et al., 2019* |
| 7 | Heart | 2520 | *Beytebiere et al., 2019* |
| 8 | NIH3T3 | 4740 | *Chiou et al., 2016* |
| 9 | Skeletal muscle | 2787 | *Dyar et al., 2018* |
| | Mean average | 3436 | |

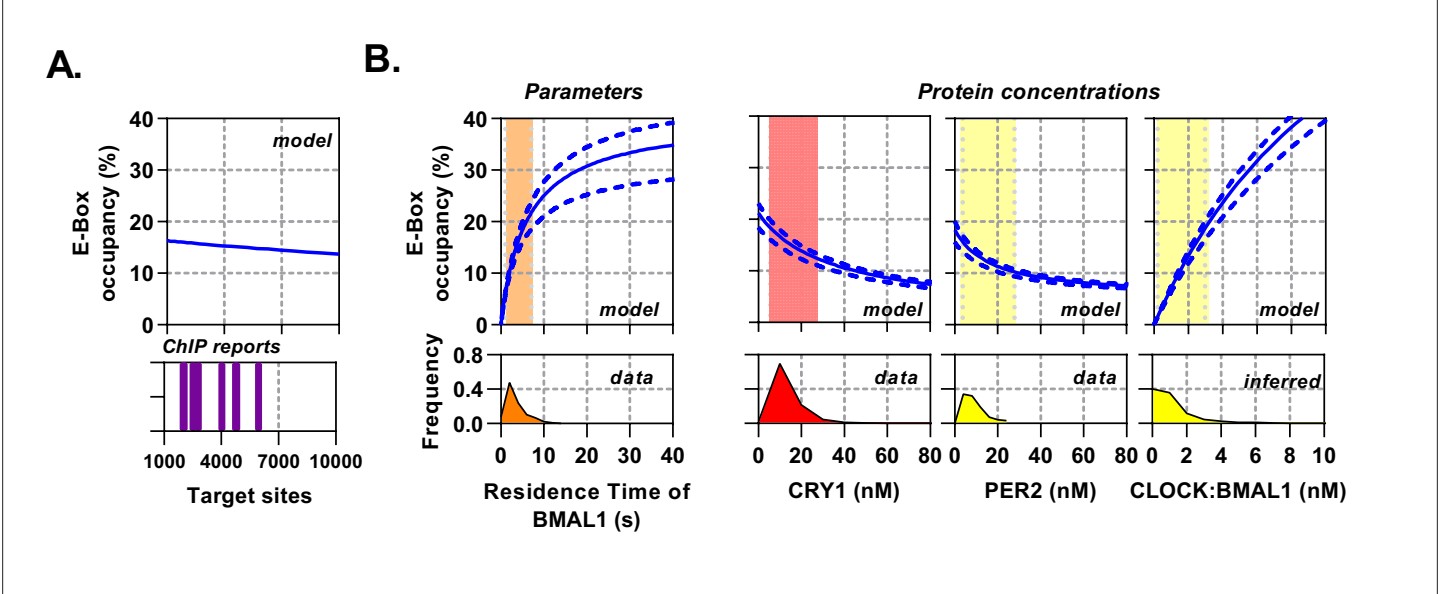

**Figure 6.** Circadian proteins operate within an optimal range to modulate E-Box binding. Sensitivity analysis of the deterministic binding model showing relationship of measured parameters (bottom) against model for occupancy of active BMAL1:CLOCK on target sites (top). (**A**) Changing number of target sites with data matched to BMAL1 ChIP data sets. (**B**) From left to right, the effect of changing residence time of CLOCK:BMAL1, or protein concentrations. Histograms show measured concentrations for corresponding proteins across all conditions/cells. The 10th to 90th percentile is highlighted. Source data available as *Figure 6—source data 1*.

The online version of this article includes the following source data for figure 6:

**Source data 1.** Model OAT outputs.

experimentally inaccessible complexes, specifically PER2:CRY1, CLOCK:BMAL1, and CLOCK:B-MAL1:CRY1 (*Figure 5D*). We find that free CLOCK:BMAL1 (unbound to DNA) cycles in remarkably low abundance from 0.9 nm to 2.4 nm, which equates to a change of ca. 809 molecules, similar to that of the PER2:CRY1 complex. Furthermore, predicted DNA binding of CLOCK:BMAL1 has an average baseline of 194 sites bound at any one time, rising to 526 sites at the peak, in agreement with the expected ≈2-fold peak enrichment from ChIP reports (*Beytebiere et al., 2019*; *Koike et al., 2012*; *Figure 5E*). The model also suggests ≈50% of the available transcription factor complex is engaged with site-specific interactions with availability predominantly limited by the $K_D$ with CLOCK. Finally, DNA-bound CLOCK:BMAL1:CRY1 complex was persistent, with low abundance cycling from 32 to 69 target sites (accounting for 11% of total BMAL1 bound sites).

## Circadian proteins are within an optimal expression range to modulate E-Box binding

The topology of the circadian molecular circuit is preserved across all cell types, yet it is also known that different cell types have widely differing repertoires of target genes and accessible genomic target sites for CLOCK:BMAL1 to bind (*Beytebiere et al., 2019*). We therefore pursued the extent to which varying the number of target sites may have an impact on the available pool of CLOCK:BMAL1 to bind target sequences, as calculated by site occupancy (the % sites occupied at any given moment). We simulated the model over a biological range of binding sites (1000 – 10,000), informed by multiple BMAL1 and CLOCK ChIP data sets (*Figure 6A*; *Beytebiere et al., 2019*; *Koike et al., 2012*). We found target site occupancy decreased marginally from 16.2% to 13.6%, showing that any variance between different numbers of available target sites has minimal impact. We then explored how varying binding parameters affected site occupancy, relating them to the variability observed in our data sets but considering values beyond these limits. The CLOCK:BMAL1 unbinding rate accounted for a 21.5% change when residence times across the observed physiological range are considered (*Figure 6B*); outside of this range occupancy begins to saturate so that a further 30 s increase in residence time only accounts for an additional 12.5% binding. Therefore, the unbinding rate, as measured

experimentally, is optimally positioned to regulate target site occupancy in a manner consistent with the displacement mechanism governed by PER:CRY.

Additionally, protein concentrations vary across circadian time as well as individual cells and cell types (*Figure 3—figure supplement 1*). We therefore simulated target-site occupancy across varied biologically plausible concentrations for CRY1, PER2, and CLOCK:BMAL1 and calculated the fraction of occupied sites (*Figure 6B*). Increasing CRY1 and PER2 led to a reduction in target-site occupancy, whereas a rise in CLOCK:BMAL1 led to a substantial increase and in both cases. Moreover, the biologically observed range occupied the most sensitive part of the curve, such that oscillations in protein copy number can evoke significant changes of occupancy. Hence, the system is positioned to make efficient use of the biological concentrations of the constituent proteins.

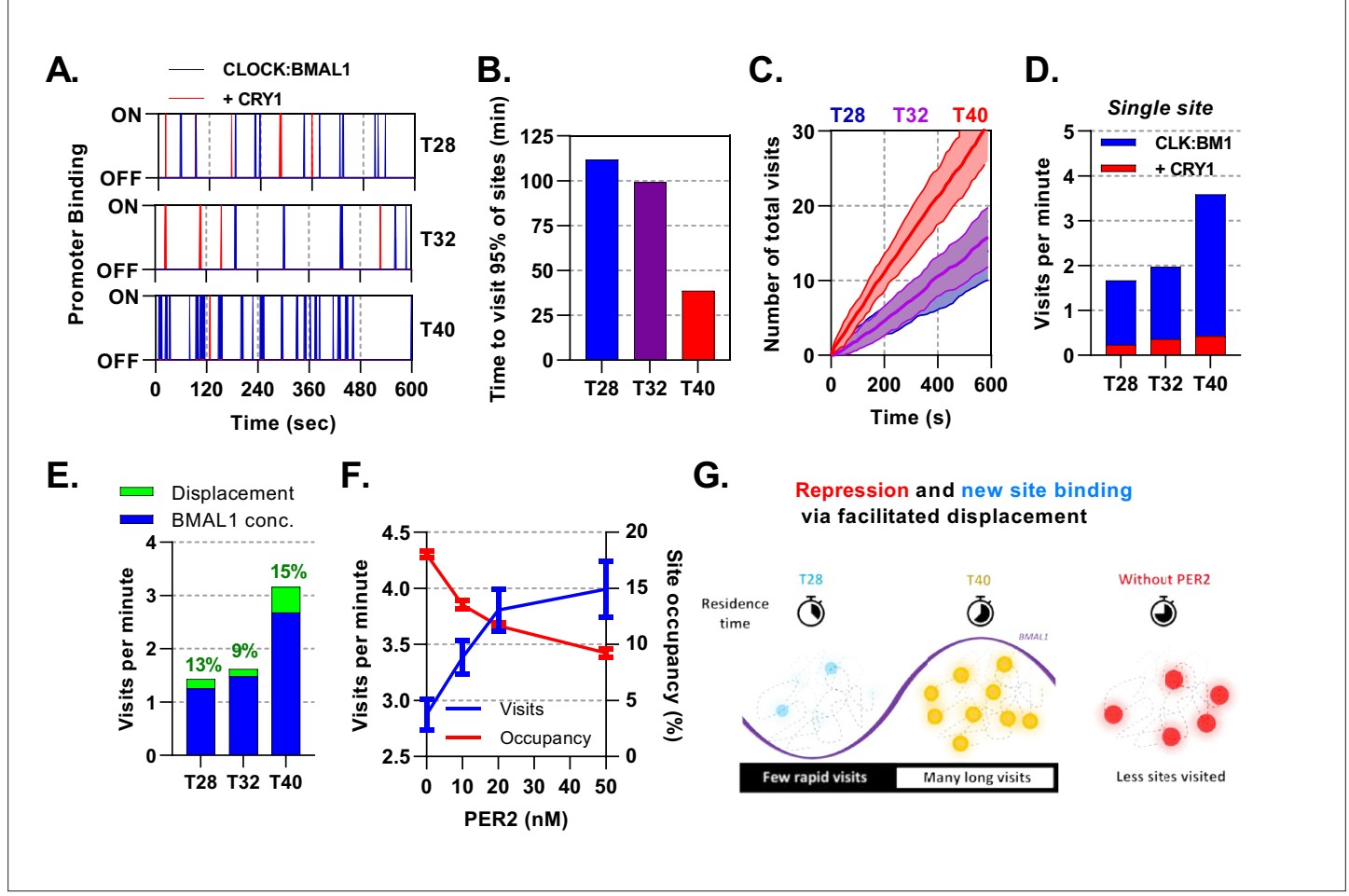

**Figure 7.** Mathematical modelling demonstrates dual function of PER:CRY mediated repression. Stochastic binding model outputs using parameters corresponding to T28, T32 or T40 post dexamethasone BMAL1 x CRY1 data sets. (**A**) Shows a promoter corresponding to the average binding rate of CLOCK:BMAL1, (**B**) the time to visit 95% of target sites once and (**C**) number of visits to a single promoter over time. Shaded error envelope shows standard deviation. (**D**) Average number of visits per minute to a target site showing active and CRY1 repressed CLOCK:BMAL1 visits. (**E**). Comparison of the contribution of BMAL1 concentration (blue) and PER2 facilitated displacement (green) on the visits per minute to a target site. Percentage contribution indicated. (**F**) Relationship of PER2 protein concentration to site visitations per minute and occupancy by CLOCK:BMAL1 using parameters for T40 explored over different concentrations of PER2. Error bars represent standard deviation. (**G**) The action of CRY:PER leads to short-lived transient binding of CLOCK:BMAL1 to DNA, working as both a repressive action whilst also facilitating binding to new target sites.

The online version of this article includes the following figure supplement(s) for figure 7:

**Figure supplement 1.** Stochastic binding model using experimentally measured parameters (**A**) Stochastic model showing the average binding (with SD) of CLOCK:BMAL1 bound target sites using input measurements from all time points for both WT and without PER2 simulations.

## Mathematical modelling demonstrates dual function of PER2:CRY1 mediated repression

Site occupancy is a function of the average residence time of transcription factors bound to DNA; consequently, highly frequent and short events would appear the same as infrequent and long binding events. To infer these masked kinetics, which are obscured in our mean based ODE model, we use a stochastic binding model to simulate individual molecules of CLOCK:BMAL1 binding to target DNA sites in a well-mixed system (*Gillespie, 1976*). For our simulations, we have used the average number of molecules and effective dissociation rates determined for T28, T32, and T40 hr post-DEX for lung fibroblasts (*Figure 7A*) arising from our previously described ODE model. T28 and T40 represent trough and peak of BMAL1 (*Figure 3D*) respectively, whereas T32 and T40 represent the trough and peak of PER:CRY protein amounts (*Figure 5D*).

Alongside binding of active CLOCK:BMAL1, we also considered target sites bound by CLOCK:B-MAL1:CRY1, which are thought to be transcriptionally inactive whilst also blocking target site access to active molecules. At T40, when there is the maximum amount of CLOCK:BMAL1, 95% of the 3,436 target sites would be bound at least once within a minute, changing to ca. 2 minutes at T28 (*Figure 7B*), contributing towards a small degree of heterogeneity. From the perspective of a single promoter at T40 there are ≈3.2 visits per minute by CLOCK:BMAL1, which is reduced down to ≈1.4 visits per minute at T28 (*Figure 7C*), with further reductions in individual cells with lower concentrations of CLOCK:BMAL1 protein (*Figure 7—figure supplement 1*). We then separated the total visits per minute into those occurring as CLOCK:BMAL1 compared to those occurring as the CLOCK:BMAL1:CRY1 complex, finding the latter to remain relatively persistent across time points and making up ≈15% of total visits, mirroring results for our ODE model. Our stochastic model therefore predicts that oscillating amounts of BMAL1 and CRY1 protein amounts, as well as the changing interaction affinity, may actually help preserve the concentration of CLOCK:B-MAL1:CRY1 target binding events across circadian time (*Figure 7D*).

Repression of CLOCK:BMAL1 activity by CRY1 requires continuous interaction and hence is limited by concentration. We hypothesised that this would be different for the PER2:CRY1-mediated displacement of CLOCK:BMAL1 from target DNA sites. To test this, we first investigated how the number of visits per minute would be affected by clamping the input values of $k_{OFF}$ and protein to different time points across different observed nuclear volumes. From this, we found that both concentration of protein and $k_{OFF}$ make a substantial contribution to the number of target site binding events (*Figure 7—figure supplement 1*). We then separated the contribution of changing amounts of CLOCK:BMAL1 protein and PER2:CRY1 mediated displacement to visits per minute by calculating the impact of removal of PER2. We find that changing BMAL1 protein abundance accounts for the most variation in number of target site visitations, changing from 1.3 visits at the nadir to 2.7 visits at peak BMAL1 protein (*Figure 7E*). CLOCK:BMAL1 mobility is supported by the action of PER2:CRY1 across all time points, accounting for a maximum 15% of visits at the trough of PER2 protein levels (T40). To explore this relationship further, we tested the impact of altering the levels of PER2 in the stochastic model, choosing four PER2 concentrations, ranging from absent to greater than observed physiological levels (0, 10, 20, and 50 nm) (*Figure 7F*). In the complete absence of PER2, BMAL1 mobility is hampered so that it visits less than three sites per minute. When PER2 spans the physiological range and beyond, a strong relationship in the visits per minute is forecast, rising by a third and in the opposite relationship to site occupancy. Our modelling demonstrates dual modes of action of PER2:CRY1, repression via displacement of CLOCK:BMAL1 from target sites and facilitation of CLOCK:BMAL1 mobility to promote new target site binding (*Figure 7G*). In this sense, PER2 acts both as part of a transcriptional repressor complex and as a facilitator of CLOCK:BMAL1 mobility to bind new target sites (*Cao et al., 2021*).

## Discussion

The circadian molecular circuit responds to and modulates an extraordinary number of biological processes, broadly imparted through DNA binding of CLOCK:BMAL1 to E-box sites (*Koike et al., 2012*). Through live cell microscopy of fluorescent ectopically and endogenously expressed circadian

proteins we have sought to understand how the autonomous molecular clock regulates CLOCK:BMAL1 binding to DNA.

## Protein abundance and stoichiometry of the circadian circuit

Mathematical modelling demonstrates that low molecular abundances, as observed for core circadian components, lead to rapid internal and cell-to-cell desynchrony, which may be compensated for by strict control of stoichiometries and interactions (*Gonze and Goldbeter, 2006*). In the first instance, protein concentrations of both activators and repressors exert significant influence on amplitude as well as robustness of daily DNA-binding cycles. We found approximately 16,000 BMAL1 and 8000 CRY1 proteins per nucleus, consistent with our earlier reports for PER2 which found 12,000 proteins per nucleus in skin fibroblasts (*Smyllie et al., 2016*). Interestingly, a recent study by Gabriel et al. found approximately an eight-fold difference between CRY1 and PER2 using the U20S, osteocarinoma cancer cell line, highlighting how different cell types and cell lines may diverge and influence the circadian network (*Gabriel et al., 2021*). Similarly, we observed significant disparities of endogenous BMAL1 across a range of cell types, with fibroblasts exhibiting a > 2 fold increase in BMAL1 concentration when compared with chondrocytes (*Figure 3—figure supplement 1*). The impact of cell type variation in protein concentrations and stoichiometries is difficult to discern but may confer tissue-specific sensitives to clock control of output genes without the need for additional regulatory components, or could compensate the system, as evidenced by similar single site visitations despite a fourfold decrease in nuclear volume (*Figure 7—figure supplement 1C*).

## Balance in affinity between BMAL1 and CLOCK may facilitate crosstalk

CLOCK was found to be cytoplasmic when ectopically expressed without BMAL1, with nuclear localisation restored upon addition of BMAL1. This suggests BMAL1 oscillations could affect availability of nuclear CLOCK, consistent with several studies (*Kondratov et al., 2003*; *Kwon et al., 2006*). Our measures of total BMAL1 and CLOCK reveal a concentration ratio of 2:1, possibly reflecting differences in turnover rate, import and export of these two proteins. Strikingly, only 10% of BMAL1 was bound in complex with CLOCK. This ratio of 2:1 is compatible with recent modelling studies defining stoichiometric relationships within the nucleus (*Lee et al., 2011*; *Kim and Forger, 2012*). Low availability of heterodimeric transcription factor for DNA interactions, when compared with free protein, severely limits the potential to bind DNA, yet this is consistent with allowing other interactions to occur, including those reported with Hypoxia-inducible factor (HIF) and Aryl hydrocarbon receptor (AhR) (*Bagnall et al., 2014*; *Jaeger and Tischkau, 2016*). Balancing availability of monomeric BMAL1 and CLOCK may therefore enable crosstalk with other pathways, or modulate interactions that have different affinities for monomeric versus heterodimeric CLOCK:BMAL1, as has been reported for CRY1 (*Michael et al., 2017*; *Xu et al., 2015*).

## Impact of cycling CRY1 concentration, binding affinities and mode of repression on the clock

Substantive evidence for direct repression of BMAL1 transactivation by CRY1 now exists (*Gustafson et al., 2017*; *Xu et al., 2015*). Here, we have shown in live cells that this interaction is not only rhythmic but remarkably strong, with a higher affinity than any other protein pairings we have measured. This strong repression of CLOCK:BMAL1 by CRY1 balances against its low abundance. When acting without PER2, CRY1 exhibits near-persistent repression over 24 hr, likely owing to its regulated interaction with CLOCK and BMAL1, as evidenced by modelling the effect of removal of either cycling BMAL1, CRY1 or binding affinity between the two (*Figure 3F*, *Figure 5—figure supplement 1F*). This cycle in affinity provides evidence that the mammalian circadian clock also relies on oscillations in the ability of key proteins to heterodimerise one another. The exact mechanisms underlying this regulation of affinity are yet to be determined but could be hypothesised to be an outcome of dimerisation with another partner that hinders or aids binding to CLOCK:BMAL1, such as PER2, or post-translational modifications leading to changes in affinity with CLOCK:BMAL1 (*Ye et al., 2011*; *Fribourgh et al., 2020*; *Schmalen et al., 2014*). A ≈25% shift in the diffusion of CRY1 equating to a change in mass close to that of, and in phase with the peak of, PER2 hints at the former proposition but further study is required (*Figures 2G and 3E*).

## Individual genes exhibit a range of residence times

We found an average short residence time of 3 s for CLOCK:BMAL1, similar to other DNA binding transcription factors including GR, p53, p65, and STAT1 (*Hettich and Gebhardt, 2018*), potentially optimised to reduce gene expression noise (*Azpeitia and Wagner, 2020*). Here we modelled CLOCK:BMAL1 binding to a number of sites using an average off rate resulting in all sites behaving the same and demonstrating how DNA binding is globally regulated, in contrast with evidence from ChiP-seq, whereby different sites are differentially bound (*Koike et al., 2012*). Presumably, robustly detected peaks found by ChIP-seq represent genes with a slow unbinding rate, such as the E-box sites found in the DBP gene, which is supported by previous live cell imaging characterising a longer 8-s residence time for BMAL1 on a DBP E-box concatemer (*Stratmann et al., 2012*). Altering the unbinding rates leads to a non-linear scaling in the occupation frequency (*Figure 5—figure supplement 1B*), highlighting the importance of regulating this parameter through post-translational modifications such as via phosphorylation of the CLOCK:BMAL1 complex as reported by *Qin et al., 2015*. Residence time may be tuned individually for different genes to ensure optimal transactivation, especially when considering recruitment of critical co-factors which do not interact with CLOCK:BMAL1 outside of DNA, as the probability of co-occupation increases with binding time. Ultimately, a considerable temporal gulf exists between the elaboration of a circadian rhythm (days) with the time-scale of DNA binding (seconds), altered by the accumulation of protein (hours). Daily changes in BMAL1 protein are moderate, remaining as high as 10,000 molecules per nucleus even at the nadir of expression, resulting in many non-transcriptionally productive interactions of CLOCK:BMAL1 with DNA throughout the circadian cycle; these interactions however may be important, contributing to pioneer factor activity and allowing others genes to activate at a different phase to BMAL1 protein levels (*Klemz et al., 2021*; *Menet et al., 2014*).

## Compromise between E-box visitations and occupancy via PER:CRY mediated displacement

Whereas CRY1 inhibits BMAL1 transactivation via binding and blocking productive interactions with transcriptional coactivators, PER:CRY complexes permit an alternative mode of repression (*Cao et al., 2021*; *Xu et al., 2015*). We demonstrate that increasing PER:CRY leads to an overall reduction in the ability for CLOCK:BMAL1 to remain bound through direct dimerisation and manipulation of DNA unbinding. Work by Cao and Wang et al revealed PER2 removes CLOCK:BMAL1 in a CRY-dependent manner from E-Boxes via recruitment of CK1 and subsequent phosphorylation of CLOCK, effectively reducing affinity for DNA (*Cao et al., 2021*). Displacive repression of this kind reduces residency time on DNA sites and thus the number of sites bound at any one time. However, reducing residency time increases the rate at which a limited pool of transcription factors can move onto new sites, hence increasing the likelihood of any one gene to be bound and reducing possible cell-to-cell variation. Site-specific residence times, most likely due to cofactor recruitment or chromatin modifications, coupled with this phenomenon would permit some gene targets to exhibit maximal CLOCK:BMAL1 binding beyond the time of the global peak. This supports findings by Menet and colleagues, who highlight groups of genes that have maximal binding events, as determined via ChIP-seq, outside of the zenith of total genome CLOCK:BMAL1 binding (*Menet et al., 2014*). Furthermore, evidence of CLOCK:BMAL1 behaving as a so-called 'kamikaze' transcription factor, a factor most transcriptionally potent when phosphorylated and marked for degradation, implies that in addition to an increase in visitations per minute, transcriptional potency is also upregulated (*Stratmann et al., 2012*). Therefore, despite the relatively high efficiency of CLOCK:BMAL1 binding to DNA, it may spend much of its life performing transcriptionally non-productive tasks until modified via complexes such as PER:CRY. PER:CRY displacement played a significant role, even at its nadir of expression, contributing to 15% of visitations per minutes at the height of DNA binding and CLOCK:BMAL1 concentration (T40). Thus, PER:CRY plays a hidden role of enhancing the mobility of CLOCK:BMAL1 to new DNA sites (*Figure 7G*).

## Materials and methods

### Key resources table

| Reagent type (species) or resource | Designation | Source or reference | Identifiers | Additional information |
|---|---|---|---|---|
| Genetic reagent (*M. musclus*) | C57BL/6 Venus::BMAL1 | *Yang et al., 2020* | | Venus sequence inserted before BMAL1 start codon. |
| Genetic reagent (*M. musclus*) | C57BL/6 Cry1::mRuby3 | This paper | | CRY1 stop codon replaced with mRuby3 |
| Genetic reagent (*M. musclus*) | C57BL/6 Venus::BMAL1 x CRY1::mRuby3 | This paper | | Crossed from Venus::BMAL1 and CRY1::mRuby3 mice |
| Cell line (*M. musculus*) | NIH/3T3 | ATCC | CRL-1658 | |
| Transfected construct (*M. musculus*) | pLNT-NLS::EGFP | Vector Builder VB900119-0501njq | | Lentiviral construct to express nuclear EGFP. |
| Transfected construct (*M. musculus*) | pLNT-BMAL1::EGFP or pLNT-BMAL1::RFP | This paper | NCBI reference: NM_007489.4 | Lentiviral construct to express fluorescent BMAL1. |
| Transfected construct (*M. musculus*) | pLNT-BMAL1-L95E::EGFP | This paper | NCBI reference: NM_007489.4 | Lentiviral construct to express fluorescent BMAL1 L95E mutant. |
| Transfected construct (*M. musculus*) | pLNT-BMAL1-V435R::EGFP | This paper | NCBI reference: NM_007489.4 | Lentiviral construct to express fluorescent BMAL1 V435R mutant. |
| Transfected construct (*M. musculus*) | pLNT-EGFP::CLOCK or pLNT-RFP::CLOCK | This paper | NCBI reference: NM_007715.6 | Lentiviral construct to express fluorescent CLOCK. |
| Transfected construct (*M. musculus*) | pLNT-EGFP::PER2 | This paper | NCBI reference: NM_011066 | Lentiviral construct to express fluorescent PER2. |
| Transfected construct (*M. musculus*) | pLNT-CRY1::RFP | This paper | NCBI reference: NM_007771.3 | Lentiviral construct to express fluorescent CRY1. |
| Chemical compound, drug | Dexamethasone | Sigma Aldrich | D4902 | |
| Software, algorithm | GraphPad Prism | GraphPad Prism | Version 9 | |
| Software, algorithm | FCCS analysis pipeline | This paper | https://github.com/LoudonLab/FcsAnalysisPipeline, (copy archived at swh:1:rev:b12e9007ed7f8a033485e57c8605e27c67df74f1; *Koch, 2021*) | |

## Plasmids

A set of lentivirus transfer plasmids encoding fluorescent fusions of circadian proteins were gener-
ated utilising the gateway cloning system as previously described *Bagnall et al., 2015*. In brief, an
initial entry vector was cloned, containing murine coding sequences for: *Bmal1* (NM_007489.4),
*Clock* (NM_007715.6), *Cry1* (NM_007771.3), and *Per2* (NM_011066.3). These vectors were then
recombined with a target destination vector containing a fluorescent protein sequence to generate
a terminal lentivirus vector, in which expression is regulated from the constitutive ubiquitin ligase C
promoter. The NLS::EGFP, BMAL1 L95E, and BMAL1 V435R encoding plasmids were all purchased
from VectorBuilder.

## Primary cell isolates and cell lines

Fibroblasts were isolated from lungs of adult mice. Lung tissue was dissected and homogenised
before collagenase-1A (1.5 mg ml$^{-1}$, Cat no. C2674) treatment for 2 hr. The cell suspension was then

filtered using 40 m cell strainers before plating into DMEM (Cat no. D6429) supplemented with 10% fetal bovine serum (HyClone), penicillin-streptomycin (10 U/ml) and amphotericin B (2.5 g ml$^{-1}$). Media was refreshed every 2–3 days for 1 week before sub-culturing or experimentation. Cells were sub-cultured for a maximum of 4 passages. SCN slice cultures were prepared as previously described (*Smyllie et al., 2016*) and imaged after 2–3 days after preparation for confocal imaging, or kept for 7 days in culture prior to bioluminescence recording. Cultures derived from separate mice were used as biological replicates.

NIH/3T3 (ATCC CRL-1658) cells were cultured in DMEM supplemented with 10% fetal bovine serum (HyClone).The cells were tested for the absence of mycoplasma using MycoAlert mycoplasma detection kit (Cat. No. LT07-418). Cells were passaged every 2–3 days, maintaining cells till passage 30. Lentivirus transduced derivatives of these cells were made using low passage cultures (6-12). Production of 3rd generation lentivirus and subsequent transduction of NIH/3T3 cells was carried out as previously described (*Bagnall et al., 2015*). Singly transduced cells are referred to with a super-script LV1 prefix before the transgene. Sequential transductions were carried out a minimum of 2 weeks later and derived cells are then termed LV2. Circadian synchronisation of cells was achieved by stimulation with 200 nm dexamethasone (Sigma D4902) for 1 hr before PBS washes and then switched to fresh culture media. Cultures were passaged for biological replicates.

## Confocal microscopy

For 2D culture imaging experiments, cells were plated into 35 mm glass bottomed imaging dishes (Greiner Bio-one) at least 6 hr prior to imaging. Measurements were performed using either a ZEISS LSM880 or ZEISS LSM780 microscope equipped with a stage mounted incubator to maintain cells at 37 °C in humidified 5% $CO_2$; fluorescence image capture was performed using either ZEN 2.1 SP3 FP2 or ZEN 2010b SP1 software, respectively. Fluorescence samples were excited using the most appropriate lasers; making use of an Argon-Ion laser to produce 488 nm or 514 nm excitation or diode laser to produce 561 nm excitation. The appropriate emitted fluorescence spectra were then collected using Quasar GasP array detectors. All images were made using a FLUAR 40 x NA 1.3 oil immersion objective. Nuclear volume recordings were made by collecting a z-stack of images at nyquist rate using a one airy unit pinhole diameter and then analysing images using Imaris (version 7.4). Time-lapse imaging of SCN: fluorescence timelapse recordings of CRY1::mRuby3 in SCN organotypic slices were acquired using Zeiss LSM780/880 inverted confocal system (Zeiss), and maintained at 37 °C. Samples were placed in air-tight glass-bottom dishes (Mattek). Images were acquired using 10 x objective, 30 s scan time per frame, 2 frames per hour, for 6 days for longer time lapse or 60–70 hr for shorter time lapse.

## Real-time population bioluminescence recordings

Lung fibroblasts were plated into 35 mm plastic tissue culture dishes (Corning). Cell media was replaced with a HEPES buffered and phenol free DMEM. Additionally, D-luciferin was supplemented into the media 4–24 hr prior to recordings. To prevent gas-exchange, dishes were sealed with grease applied around the edges of the coverslips. Bioluminescence was then recorded by photomultiplier tubes (PMTs; Hamatasu) housed in an enclosed incubator at 37 °C and without $CO_2$ as described previously (*Loudon et al., 2007*).

## Single-molecule fluorescence in situ hybridisation

*Clock* and *Bmal1* mRNA were visualised using custom probes designed against *Clock* and *Bmal1* murine coding sequences via the Stellaris FISH Probe Designer (Biosearch Technologies Inc). *Clock* and *Bmal1* probes were labeled with the Quasar-570 and Quasar-670 dyes, respectively. Samples were imaged with a wide-field DeltaVision microscope as previously described and spot counting was performed with FISH-quant (*Bagnall et al., 2018*; *Mueller et al., 2013*).

## Fluorescence recovery after photo-bleaching

FRAP was performed by time-lapse imaging of cells prior to and after photobleaching to visualise fluorescence recovery. Photobleaching of EGFP and Venus signals was performed using 488 nm or 514 nm laser lines respectively using circular regions of 5 m diameter (approximately 10% nuclear area) and wholly within the nuclei of cells. Images were recorded every 0.262 s for up to 60 s.

We have used FRAP to infer DNA unbinding rates for BMAL1, see *Figure 1—figure supplement 2*. The principle of the approach assumes that a combination of diffusion and binding to an unseen immobile substrate affects the speed in which fluorescent proteins move into the recovery area. Different trajectories of recovery therefore inform how the balance between binding and diffusion contributes to the apparent diffusion of the observed molecule. This approach has been utilised many times to characterise the binding times of transcription factors to DNA, including GR, STAT1, p53, and p65 and has been additionally cross validated against single molecule imaging (*Groeneweg et al., 2014*). For our data, the recovery curves of BMAL1::EGFP (co-expressed alongside tagRFP::CLOCK) remained consistent when bleaching different sized nuclear regions indicating that binding contributes to the recovery profile rather than a solely diffusion-led process (*Figure 1—figure supplement 2*). For all subsequent measurements, a circular bleach region was used that was kept consistent across cells and accounted for approximately 10% of nuclear area. FRAP was performed and analysed using the appropriate ZEN software version with recovery curves from the bleached region normalised to total cell fluorescence as well as background fluorescence (empty spaces away from cells). The normalised recovery curve of fluorescence within the bleached region over time, $t$, was then fit with a reaction binding model

$$f(t) = I_E - I_1 e^{-\frac{t}{\tau}}, \tag{1}$$

where $\tau$ is the residence time (reciprocal of unbinding rate $k_{\text{OFF}}$), $I_E$ and $I_1$ is the immobile and mobile fraction, respectively (*Sprague and McNally, 2005*).

## Fluorescence correlation spectroscopy

### Experimental setup
FCS measurements were performed in each cell nucleus using acquisition times of 20 s and a collection volume of 1 airy unit (approximately 0.722 fl and 1.10 fl for 488 nm and 561 nm excitation respectively when using the FLUAR 40 x NA 1.3 oil immersion objective) calibrated in the x-y plane for maximum signal intensity. The effective confocal volumes were calculated via the equation

$$V_{\text{eff}} = (2\pi)^{\frac{3}{2}} w_{xy}^2 w_z = (2\pi)^{\frac{3}{2}} \left( \frac{0.61\lambda}{\text{NA}} \right)^2 \left( \frac{2n\lambda}{\text{NA}^2} \right), \tag{2}$$

where $w_{xy}$ and $w_z$ is the beam width in the $xy$ plane and $z$ axis respectively, with NA as the numerical aperture (NA = 1.3 for our 40 x objective), $\lambda$ the wavelength of exciting laser and $n$ the refractive index of the immersion oil ($n = 1.515$ in all experiments). The appropriate spectra were collected for each different fluorophore. Laser power was reduced to minimise photo-bleaching whilst maintaining counts per molecule greater than 0.3 kHz.

### Fitting
Auto-correlation curves extracted from the Zeiss.fcs files were fit over two rounds using a program written in Python 3; first a global parameter fit executed using a genetic algorithm *differential evolution* (SciPy [*Virtanen et al., 2020*]) generating initial guesses within reasonable parameter bounds was performed, followed by a final stage of non-linear least-squares regression implemented via the *curve fit* (SciPy) package with an arctan loss function. The non-linear regression was regularised using the standard deviation following the calculations by *Saffarian and Elson, 2003* which incorporates systematic sources of error at short and long lag times due to the multi-tau correlation algorithm used to compute the correlation curve; at short lag times the averaging introduces uncertainty whilst at the long lag times less data points exist to correlate due to the finite time over which the experiment was run. Poor fits arising from samples expressing only auto-fluorescence or no fluorescence from focus shifts can result in highly non-plausible parameter measurements (diffusion >100 um/s) and concentrations ( > 1000 nM) which were removed by robust regression and outlier removal (ROUT) (*Motulsky and Brown, 2006*).

### Model selection
The Akaike Information Criterion (AIC) (*Akaike, 1974*) was used to score and select the best fit model with the lowest score, defined as

$$AIC = 2k - 2\ln(\hat{L}), \tag{3}$$

where $k$ is the number of fitted parameters and $\hat{L}$ the maximum likelihood, equal to the sum of squared errors when using non-linear least squares regression to fit the curves. Results of the model selection for all FCS data sets performed in this study can be found in *Figure 2—figure supplement 1*.

## Interactions: fluorescence cross-correlation spectroscopy

Care was taken for fluorescence cross-correlation spectroscopy (FCCS) measurements to avoid the green channel signal spilling up into the red channel causing false cross-correlation by reducing the laser power and observing the far-red part of the second channel. Control measurements were performed by selectively turning off either 488 or 561 nm lasers and tuning the red channel spectra until there was no cross-correlation due to spill-over. We analysed both sets of auto-correlation and cross-correlation curves from the same measurement and used the same procedure as *Sadaie et al., 2014* to calculate the disassociation constant $K_D$. We again used non-linear least squares regression upon this data, fitting the function

$$\frac{[\text{Complex}]}{[\text{A::GFP}]_{\text{TOTAL}}} = \frac{[\text{B::RFP}]_{\text{TOTAL}} - [\text{Complex}]}{K_D + [\text{B::RFP}]_{\text{TOTAL}} - [\text{Complex}]}, \tag{4}$$

where $[\text{A::GFP}]_{\text{TOTAL}}$ is the total concentration of the protein $A$ fused to a green or yellow fluorescent protein, $[\text{B::RFP}]_{\text{TOTAL}}$ is the total of protein $B$ fused to a red fluorophore and $[\text{Complex}]$ is the concentration of the dimer of $A$ and $B$ proteins. The standard deviation upon $K_D$ was also provided by this algorithm.

## Maturation correction

Fluorescent proteins may take minutes or hours to fold correctly before becoming visible, with the invisible fraction becoming substantial if the degradation rate of the protein is comparable to the maturation rate, hence leading to misreports in protein number as measured by FCS. The red fluorescent protein, mRuby3 is known to have a long maturation time of 2.28 (*Balleza et al., 2018*) and CRY1 to have a half-life of approximately 2.1 (*Yoo et al., 2013*), therefore we applied a scaling correction to CRY1::mRuby3 FCS concentration data. To account for the unseen portion, we model the protein in two states; an invisible state, $P$, and a mature visible fraction, $M$. Assuming a constant rate of production, $k_p$, for the immature protein, a maturation rate for the fluorophore of $k_m$, and a degradation rate for both protein states of $k_d$ we get the set of ordinary differential equations

$$\begin{aligned} \frac{dP}{dt} &= k_p - k_d P - k_m P, \\ \frac{dM}{dt} &= k_m P - k_d M. \end{aligned} \tag{5}$$

These equations may be solved analytically using an integrating factor assuming zero of both protein states at $t = 0$ and so long as the rate constants $k_m$ and $k_d$ are greater than zero. The unknown production rate, $k_p$, is divided out when computing the ratio of both states by $M$ and taking the limit of the solution as $t \longrightarrow \infty$ to yield the correction factor

$$c = \lim_{t \to \infty} \left( \frac{P(t) + M(t)}{M(t)} \right) = \frac{k_d + k_m}{k_m} = \frac{\tau_m}{\tau_d} + 1, \tag{6}$$

where $\tau_m$ and $\tau_d$ are the doubling-time and half-life of the maturation and degradation respectively. Using *Equation (6)*, the half-life for CRY1 and the maturation time of mRuby3 we find a multiplicative factor of $c = 2.083$, which may multiply the observed protein to yield the total concentration of CRY1::mRuby3.

## Diffusion rate as a function of mass

When considering normal diffusion due to Brownian motion the diffusion rate, $D$, may be modelled using the Stokes–Einstein equation (*Einstein, 1905*)

$$D = \frac{k_B T}{8\pi\eta r}, \tag{7}$$

where $k_B$ is the Boltzmann constant, $T$ the temperature in kelvin, $\eta$ the dynamic viscosity, and $r$ as the radius of the diffusing molecule. Assuming a constant density of spatially equally distributed constituent amino acids, the mass of the molecule grows like $r^3$ and hence the diffusion rate will be related to the mass of the molecule by

$$D \propto m^{-1/3}, \tag{8}$$

hence a halving in mass will equate to an approximate increase of 1.26 times the diffusion rate.

## In vitro binding assays

### Expression and purification of recombinant proteins

Biotin Acceptor Peptide (BAP)-tagged CLOCK PAS-AB (mouse CLOCK residues 93–395) was expressed as a His$_6$-NusA-XL-tagged protein in *Escherichia coli* (*E. coli*) Rosetta2 (DE3) cells. The *E. coli* biotin ligase BirA was expressed as a GST-tagged protein in BL21 (DE3) cells. Protein expression was induced with 0.5 mM isopropyl-$\beta$-D-thiogalactopyranoside (IPTG) at an OD$_{600}$ of approximately 0.8 and grown for an additional 16 hr at 18 °C. Cells were centrifuged at 4 °C at 3200 x g, resuspended in 50 mM Tris pH 7.5, 300 mM NaCl, 5% (vol/vol) glycerol, and 5 mM-mercaptoethanol (BME) and lysed using a microfluidizer followed by brief sonication on ice. After clarifying lysate by centrifugation at 4 °C at 140,500 x g for 1 hr, proteins were captured using Ni-NTA resin (Qiagen) or Glutathione Sepharose 4B resin (GE Life Sciences). After extensive washing in in 50 mM Tris pH 7.5, 300 mM NaCl, 5% (vol/vol) glycerol, and 5 mmM BME, the affinity and solubility tags (e.g. His$_6$-NusA-XL or GST) were cleaved on resin using GST-TEV or His6-TEV protease at 4 °C overnight. Cleaved proteins were collected from the flow-through; GST-BirA was further purified using size exclusion chromatography (SEC) on a Superdex75 column (GE Healthcare) in 50 mM Tris, pH 8.0, 300 mM NaCl, 1 mM dithiothreitol (DTT), and 5% (vol/vol) glycerol, while CLOCK PAS-AB was further purified using SEC in 20 mM HEPES pH 7.5, 125 mM NaCl, 5% (vol/vol) glycerol, and 2 mM Tris(2-carboxyethyl)phosphine (TCEP).

BMAL1 PAS-AB (mouse BMAL1 residues 136–441) was expressed in Sf9 suspension insect cells (Expression systems) as a GST-tagged protein using the baculovirus expression system. Sf9 cells were infected with P3 virus at $1.2 \times 10^6$ cells per milliliter and grown for 72 hr at 27 °C before harvesting. Cells were resuspended in resuspension buffer (50 mM HEPES pH 7.5, 300 mM NaCl, 5% (vol/vol) glycerol, and 5 mM-mercaptoethanol (BME)). Cells were lysed using a microfluidizer followed by brief sonication on ice. After clarifying lysate by centrifugation at 140,500 x g for 1 hr at 4 °C, the lysate was bound in batch-mode to Glutathione Sepharose 4B resin (GE Healthcare), washed in resuspension buffer and eluted with 50 mM HEPES pH 7.5, 150 mM NaCl, 5% (vol/vol) glycerol, 5 mM BME, and 25 mM reduced glutathione. The protein was desalted into 50 mM HEPES pH 7, 150 mm NaCl, 5% (vol/vol) glycerol, and 5 mM BME using a HiTrap Desalting column (GE Healthcare) and incubated with GST-TEV protease overnight at 4 °C. The cleaved GST-tag and GST-tagged TEV protease were removed by Glutathione Sepharose 4B chromatography (GE Healthcare) and the BMAL1 PAS-AB was further purified by SEC on a Superdex75 column (GE Healthcare) in 20 mm HEPES pH 7.5, 125 mM NaCl, 5% (vol/vol) glycerol, and 2 mM TCEP. For long-term storage, small aliquots of proteins were frozen in liquid nitrogen and stored at –70 °C.

Biotinylation of CLOCK PAS-AB. For the biotinylation reaction, 100 m BAP-CLOCK PAS-AB was incubated in 20 mM HEPES pH 7.5, 125 mM NaCl, 5% (vol/vol) glycerol, and 2 mM TCEP with 2 mM ATP, 1 m GST-BirA, and 150 m biotin at 4 °C overnight. GST-BirA was removed after the reaction using Glutathione Sepharose 4B resin (GE Healthcare) and excess biotin was separated from the labeled protein by SEC on a Superdex75 column in 20 mm HEPES pH 7.5, 125 mM NaCl, 5% (vol/vol) glycerol, and 2 mM Tris(2-carboxyethyl)phosphine (TCEP). Biotinylation of CLOCK PAS-AB was essentially complete, as determined by incubating the protein with excess streptavidin and resolving complexes on SDS-PAGE. For long-term storage, small aliquots of the biotinylated protein were frozen in liquid nitrogen and stored at –70 °C.

### Surface plasmon resonance binding assays

Kinetic binding experiments were conducted on a Biacore X100 +instrument (GE Healthcare) capturing biotinylated CLOCK PAS-AB on a streptavidin-coated SA sensor chip at 100–150 Response Units (RUs). Serial dilutions of BMAL1 PAS-AB from 0.25 to 10 nm were injected in phosphate buffered saline (PBS) over 250 s and dissociated into buffer over 250 s to determine binding kinetics.

Sensorgram data were globally fit to a 1:1 biomolecular binding model with Biacore Evaluation software X100 +version 2.0.1 (GE Healthcare) to determine $k_{ON}$, $k_{OFF}$ and $K_D$. $\chi^2$ values $lt_1$ and $R_{max} \leq 100$ were established as quality cutoffs for acceptable data. See *Figure 2—figure supplement 2F* for surface plasmon resonance results between BMAL1 PAS-AB and CLOCK PAS-AB domains.

## Animal lines

A previously established Venus::BMAL1 mouse line was used (*Yang et al., 2020*). Additionally, two more mouse lines were generated which included CRY1::mRuby3 (*Figure 3—figure supplement 1*) and a subsequent cross with mice expressing Venus::BMAL1 and PER2::LUC (*Bagnall et al., 2015*). The CRY1::mRuby3 mice were made using a CRISPR-mediated genomic editing approach to introduce a fluorescent sequence via homology-directed repair. Details of methodology and validation of animals can be found in the supplementary materials. Eight- to 10-week-old mice were housed in individual cages in light-tight cabinets (Tecniplast), equipped with activity mouse wheel cages (Actimetrics). Activity was recorded by ClockLab data collection software in 6-min bins (Actimetrics). The mice were maintained at LD cycles (light on at 7 am; light off at 7 pm) for 2 weeks. Activity profiles were generated using ClockLab (Actimetrics) and used to apply Non-Parametric Circadian Rhythm Analysis (NPCRA) to 10 circadian days of wheel-running data, as described previously (*Reppert and Weaver, 2002*), to calculate: Intra-daily Variability (IV): Non-parametric frequency of activity-rest transitions within a day, with a range of between 0 and 2 (e.g. a Sine wave would have a value of 0 and Gaussian noise would have a value of 2). Inter-daily Stability (IS): Matching of activity patterns on day-to-day basis, ranging from 0 (Gaussian noise) to 1 (high-stability). Robust behavioural activity is characterised by low IV and high IS. ClockLab (Actimetrics) was used to generate double-plotted actograms with onsets of activity and phase angle of entrainment was calculated from 10 days of wheel-running data measuring the difference in time of the point in the entraining cycle (lights on) against the onset of activity.

## Generation of CRY1::mRuby3 mouse line

We used CRISPR-Cas9 to generate C terminally tagged alleles for Cry1, see *Figure 3—figure supplement 2*. Two sgRNA targeting the STOP codon of the gene were selected using the Sanger WTSI website (*Hodgkins et al., 2015*) that adhered to our criteria for off target predictions (guides with mismatch (MM) of 0, 1 or 2 for elsewhere in the genome were discounted, and MM3 were tolerated if predicted off targets were not exonic). sgRNA sequences, wih PAM site indicated in italics, (aactgata cggtaaatactt-*AGG* and cggcagagcagtaactgata-*CGG*) were purchased as crRNA oligos, which were annealed with tracrRNA (both oligos supplied by IDT) in sterile, RNase free injection buffer (TrisHCl 1 mM, pH 7.5, EDTA 0.1 mM) by combining 2.5 mg crRNA with 5 mg tracrRNA and heating to 95°C, which was allowed to slowly cool to room temperature.

For our donor repair template, we used the EASI-CRISPR long-ssDNA strategy (*Quadros et al., 2017*), which comprised of the mRuby3 gene with linker flanked by 132 and 143 nt homology arms synthesised by a Biotinylation PCR and on-column denaturation method (*Bennett et al., 2021*; *Figure 3—figure supplement 3A*). For embryo microinjection, the annealed sgRNA was complexed with Cas9 protein (New England Biolabs) at room temperature for 10 min, before addition of long ssDNA donor (final concentrations; sgRNA 20 ng/ml, Cas9 protein 20 ng/ml, lssDNA 10 ng/ml). CRISPR reagents were directly microinjected into C57BL6/J (Envigo) zygote pronuclei using standard protocols. Zygotes were cultured overnight and the resulting two-cell embryos surgically implanted into the oviduct of day 0.5 post-coitum pseudopregnant mice. Potential founder mice were screened by PCR (*Figure 3—figure supplement 2A*), using primers that flank the sgRNA sites (Cut test F taca ctatgctcacgggggac and Cut test R accacgtcctcttcagaacc), which both identifies editing activity in the form of InDels from NHEJ repair, and can also detect larger products indicating HDR (*Figure 3—figure supplement 2A*). Pups 18, 19, and 22, which gave positive products in PCR reactions, were sequenced by amplifying again with the cut test F/R primers using high fidelity Phusion polymerase (NEB), gel extracted and subcloned into pCRblunt (Invitrogen) and Sanger sequenced with M13 Forward and Reverse primers. All pups showed perfect sequence integration and were bred with a WT C57BL6/J to confirm germline transmission.

## Mathematical modelling

### Modelling aims and assumptions

We sought to model how the core circadian transcription factor, CLOCK:BMAL1, binds to specific E-BOX DNA sites over daily cycles in concentration and interactions with the key repressors CRYP-TOCHROME1 (CRY1) and PERIOD2 (PER2). We have opted to neglect explicitly modelling the non-specific DNA interactions, such as sliding, hopping and intersegmental transfer, as we have no direct measurements of these properties. Instead we chose to allow the specific site on rate ($k_{ON}$) to represent all protein-DNA processes required to achieve binding to an specific site by fitting $k_{ON}$ alongside other ON rates. CRY1 and PER2 repress the activity of BMAL1 through direct binding of the trans-activation domain (TAD) to block transcriptional potential and the promotion of weaker binding to DNA, respectively. We have assumed that PER2 may interact with BMAL1 and CLOCK:BMAL1 only via CRY1 with the same affinity that CRY1 alone has for BMAL1. To constrain the on rates during fitting, we have used the measured disassociation constants, $K_D$, between CLOCK-BMAL1 (*Figure 2G*), CLOCK:BMAL1-EBOX (*Huang et al., 2012*), CRY1-PER2 (*Figure 4C*), and the rhythmic CRY1-BMAL1 $K_D$ (*Figure 3F*). All protein-protein and protein-DNA interactions are modelled as explicit dimerisation events leading to a new species dependent on an ON and OFF rate. A summary of the parameters, $K_D$ values and which parameters were proposed during fitting is given in *Table 1*. Following the convention when defining chemical reactions, square brackets are used to signify concentrations of the species within. To aid understanding of the reactions being modelled we describe the species participating in reactions as familiar initialisations, for example [CB] represents the concentration of the CLOCK:BMAL1 heterodimer and [C1] for CRY1. Consequently, further dimerisations or bound states are denoted by concatenations of these initialisations, for example [CBC1P2] for the CLOCK:B-MAL1:CRY1:PER2 tetramer or [CBS] for CLOCK:BMAL1 bound to a specific DNA ($S$) site.

### Dimerisation

Hetero-dimerisation of two species [A] and [B] proceeds to the dimer [AB] via the reaction

$$[A] + [B] \rightleftharpoons^{k_{ON}}_{k_{OFF}} [AB], \tag{9}$$

where where $k_{ON}$ ($nm^{-1} s^{-1}$) and $k_{OFF}$ ($s^{-1}$) are the forward and backwards rates respectively (*Sadaie et al., 2014*), often referred to as the association and dissociation rate constants. In equilibrium, the forward rate of reaction is equal to the backward rate resulting in the definition of the disassociation constant

$$K_D = \frac{[A][B]}{[AB]}, \tag{10}$$

defined in terms of the ON and OFF rates as

$$K_D = \frac{k_{OFF}}{k_{ON}}. \tag{11}$$

A stronger interaction is represented as a smaller $K_D$ value as the rate to the disassociated is smaller than the association rate. In the limit of long times, $t \to \infty$, the concentration of the dimer [AB] in equilibrium becomes

$$[AB]_{eq} = \frac{[A]_0 + [B]_0 + K_D - \sqrt{([A]_0 + [B]_0 + K_d)^2 - 4[A]_0[B]_0}}{2}, \tag{12}$$

where a subscript 0 denotes the initial concentration. Alternatively, assuming no production or degradation terms exist we may simulate analytically intractable multiple interactions by simulating a coupled ODE model until equilibrium concentrations are reached. For all ODE modelling, we defined equilibrium as less than a 1% deviation in molecular concentrations over the last 20% of simulated time points. In all cases equilibrium was established in less than 30 min of simulated time, smaller than the window over which experimental FCS time point measurements were performed.

## Ordinary differential equation model of DNA binding

Systems of ordinary differential equations (ODE), modelling the concentrations of molecular species, were solved in the Python three programming language to reflect measured interactions between

different molecules and DNA. All interactions are modelled as explicit dimerisation events yielding a new molecular species. ODEs were solved in time as an initial value problem using the LSODA solver as implemented in the SciPy *odeint* function (**Virtanen et al., 2020**) and ran until equilibrium concentrations were reached, typically reached in less than 30 min of simulated time. The system of ODEs are

$$\frac{d[CB]}{dt} = -d_{ON}[CB][C1P2]+d_{OFF}[CBC1P2]-d_{ON}[CB][C1]+d_{OFF}[CBC1]-k_{ON}[CB][S] +k_{OFF}[CBS]+b_{ON}[C][B]-b_{OFF}[CB],$$

$$\frac{d[S]}{dt} = -k_{ON}[CB][S] + k_{OFF}[CBS] - k_{ON}[CBC1P2][S] + R_{OFF}[CBC1P2S] - k_{ON}[CBC1][S] + k_{OFF}[CBC1S],$$

$$\frac{d[CBS]}{dt} = k_{ON}[CB][S] - k_{OFF}[CBS] - d_{ON}[CBS][C1] + d_{OFF}[CBC1S] - d_{ON}[CBS][C1P2] + d_{OFF}[CBC1P2S],$$

$$\frac{d[C1]}{dt} = -d_{ON}[CB][C1]+d_{OFF}[CBC1]-d_{ON}[CBS][C1]+d_{OFF}[CBC1S]-a_{ON}[C1][P2] +a_{OFF}[C1P2]-d_{ON}[B][C1] + d_{OFF}[BC1],$$

$$\frac{d[CBC1]}{dt} = -a_{ON}[CBC1][P2] + a_{OFF}[CBC1P2] + d_{ON}[CB][C1] - d_{OFF}[CBC1] - k_{ON}[CBC1][S] + k_{OFF}[CBC1S] + b_{ON}[C][BC1] - b_{OFF}[CBC1],$$

$$\frac{d[CBC1S]}{dt} = d_{ON}[CBS][C1] - d_{OFF}[CBC1S] - a_{ON}[CBC1S][P2] + a_{OFF}[CBC1P2S] + k_{ON}[CBC1][S] - k_{OFF}[CBC1S],$$

$$\frac{d[P2]}{dt} = -a_{ON}[CBC1][P2] + a_{OFF}[CBC1P2] - a_{ON}[C1][P2] + a_{OFF}[C1P2] - a_{ON}[CBC1S][P2] + a_{OFF}[CBC1P2S] - a_{ON}[BC1][P2] + a_{OFF}[BC1P2],$$

$$\frac{d[C1P2]}{dt} = -d_{ON}[CB][C1P2] + d_{OFF}[CBC1P2] + a_{ON}[C1][P2] - a_{OFF}[C1P2] - d_{ON}[CBS][C1P2] + d_{OFF}[CBC1P2S] - d_{ON}[B][C1P2] + d_{OFF}[BC1P2],$$

$$\frac{d[CBC1P2S]}{dt} = k_{ON}[CBC1P2][S] - R_{OFF}[CBC1P2S] + a_{ON}[CBC1S][P2] - a_{OFF}[CBC1P2S] + d_{ON}[CBS][C1P2] - d_{OFF}[CBC1P2S],$$

$$\frac{d[CBC1P2]}{dt} = a_{ON}[CBC1][P2] - a_{OFF}[CBC1P2] + d_{ON}[CB][C1P2] - d_{OFF}[CBC1P2] - k_{ON}[CBC1P2][S] + R_{OFF}[CBC1P2S] + b_{ON}[C][BC1P2] - b_{OFF}[CBC1P2],$$

$$\frac{d[C]}{dt} = -b_{ON}[C][B] + b_{OFF}[CB] - b_{ON}[C][BC1] + b_{OFF}[CBC1] - b_{ON}[C][BC1P2] + b_{OFF}[CBC1P2],$$

$$\frac{d[B]}{dt} = -b_{ON}[C][B] + b_{OFF}[CB] - d_{ON}[B][C1] + d_{OFF}[BC1] - d_{ON}[B][C1P2] + d_{OFF}[BC1P2],$$

$$\frac{d[BC1]}{dt} = d_{ON}[B][C1] - d_{OFF}[BC1] - b_{ON}[C][BC1] + b_{OFF}[CBC1] - a_{ON}BC1[P2] + a_{OFF}[BC1P2],$$

$$\frac{d[BC1P2]}{dt} = d_{ON}[B][C1P2] - d_{OFF}[BC1P2] - b_{ON}[C][BC1P2] + b_{OFF}[CBC1P2] + a_{ON}[BC1][P2] - a_{OFF}[BC1P2].$$

A genetic algorithm, implemented in *differential evolution* (SciPy [**Virtanen et al., 2020**]), was utilised to fit the unknown parameters in the ODE model via Chi-squared minimisation to experimental $k_{OFF}$ mean and standard error on the mean using an in silico value, $\bar{k}_{OFF}$, generated by the model. All non-dimerised species concentrations, as measured experimentally, were introduced for each of the seven time-points – a 24 hr time span sampled every 4 hr – into the model as inputs alongside measured disassociation constants to constrain fitted OFF rates as a function of proposed ON rates, reducing the number of fitted parameters. A summary of the parameters in the model is given in *Table 1*. During fitting the in silico $k_{OFF}$ value was calculated by allowing all species to reach equilibrium after setting all DNA bound species to zero following an initial run of the model, the resultant equilibrium concentrations of bound and free molecules were used to calculate the off rate (*Figure 5—figure supplement 1A*). The average apparent DNA unbinding rate $\bar{k}_{OFF}$, which is analogous to the same rate as experimentally measured in FRAP, is simulated following the method by **Röding et al., 2019** through rearranging *Equation (11)* for the off rate

$$\bar{k}_{OFF} = k_{ON}K_D = k_{ON}\frac{[\text{Unbound CB}][\text{Unbound Sites}]}{[\text{Bound Sites}]}, \tag{13}$$

with

$$[\text{Unbound CB}] = [CB]_{eq} + [CBC1]_{eq} + [CBC1P2]_{eq}, \tag{14}$$

$$[\text{Unbound Sites}] = [S]_{eq}, \tag{15}$$

$$[\text{Bound sites}] = [CBS]_{eq} + [CBC1S]_{eq} + [CBC1P2S]_{eq} \tag{16}$$

where eq denotes concentrations at equilibrium as $t \to \infty$. The apparent $\bar{k}_{OFF}$ is an average of the CLOCK:BMAL1-DNA binding OFF rate $k_{OFF}$ and CLOCK:BMAL1:CRY1:PER2-DNA OFF rate $R_{OFF}$ weighted by their respective relative concentrations, with increasing levels of CRY1:PER2 increasing $\bar{k}_{OFF}$ as $k_{OFF} < R_{OFF}$. The fitted parameters are given in *Table 1* and predicted in silico $\bar{k}_{OFF}$ values for WT and PER2 KO can be seen in (*Figure 5—figure supplement 1C*). Knocking out PER2 (keeping all other species and parameters the same as wild type values) removes all rhythmic regulation of $\bar{k}_{OFF}$ and ensures that CLOCK:BMAL1 is bound for longer at all time points such that the number of bound specific sites (S) also increases for all time points (*Figure 5—figure supplement 1E*). Locking BMAL1, CRY1, PER2 and the interaction between BMAL1 and CRY1 to their mean value between 24 and 48 hr post-dexamethosone (DEX) treatment, reveals that setting BMAL1 to its mean value significantly alters both bound and free from DNA CLOCK:BMAL1:CRY1 whilst locking the other rhythmic components has little impact (*Figure 5—figure supplement 1F*).

Upper and lower bounds on one-at-a-time (OAT) sensitivity analysis (*Figure 6A–B*) were generated by running the fitted model for both an estimate of the smallest and largest number of target sites, 1000 and 10,000 respectively, with the mean representing the mean number of target sites from ChIP data, namely 3,436. We may estimate the number of target sites for CLOCK:BMAL1 from previous studies investigating high confidence sites that BMAL1 binds to in ChIP-seq, with *Table 2* outlining the reports and peaks measured via ChIP-seq that were used in estimating the number of target sites used in our mathematical modelling. In addition to the OAT analyses in *Figure 6A–B* we also examined how changing amounts of CRY1:PER2 alters the residence time of CLOCK:BMAL1 on DNA, demonstrating how CRY1:PER2 readily promotes removal from DNA in a non-linear fashion over a physiologically plausible range of concentrations (*Figure 5—figure supplement 1B*).

## Stochastic DNA binding model

Stochastic binding simulations in Python three utilised the Gillespie algorithm (*Gillespie, 2002*) through the StochPy library (*Maarleveld et al., 2013*) to simulate a reduced topology, considering

**Table 3.** Stochastic model reactions and propensities.

Counter for arrivals by CLOCK:BMAL1 ($CB$) without CRY1 ($C1$) to previously unbound sites $S$ converting them to $S_0$ given by $A_{CB}$ as well as counters for marked site $M$ binding represented by $B_X$, and unbinding, $U_X$, by species $X$. The size of the system is given by $\Omega = 1 \times 10^{-9} N_A V$, where $V$ is the volume in liters and is used to convert ON rate quantities with dimensions nm⁻¹s⁻¹ into particle⁻¹ s⁻¹. $k_{ON}$ is the same value as previously fitted for the ODE model given in *Table 1*.

| No. | Reaction | Propensity |
|---|---|---|
| 1 | $CB + S \longrightarrow CBS + A_{CB}$ | $(k_{ON}/\Omega) \cdot CB \cdot S$ |
| 2 | $CB + S_0 \longrightarrow CBS$ | $(k_{ON}/\Omega) \cdot CB \cdot S_0$ |
| 3 | $CBS \longrightarrow CB + S_0$ | $k_{OFF} \cdot CBS$ |
| 4 | $CBC1 + S \longrightarrow CBC1S$ | $(k_{ON}/\Omega) \cdot CBC1 \cdot S$ |
| 5 | $CBC1S \longrightarrow CBC1 + S$ | $k_{OFF} \cdot CBC1S$ |
| 6 | $CBC1 + S_0 \longrightarrow CBC1S_0$ | $(k_{ON}/\Omega) \cdot CBC1 \cdot S_0$ |
| 7 | $CBC1S_0 \longrightarrow CBC1 + S_0$ | $k_{OFF} \cdot CBC1S_0$ |
| 8 | $CB + M \longrightarrow CBM + B_{CB}$ | $(k_{ON}/\Omega) \cdot CB \cdot M$ |
| 9 | $CBM \longrightarrow CB + M + U_{CB}$ | $k_{OFF} \cdot CBM$ |
| 10 | $CBC1 + M \longrightarrow CBC1M + B_{CBC1}$ | $(k_{ON}/\Omega) \cdot CBC1 \cdot M$ |
| 11 | $CBC1M \longrightarrow CBC1 + M + U_{CBC1}$ | $k_{OFF} \cdot CBC1M$ |

only CLOCK:BMAL1 and CLOCK:BMAL1:CRY1 binding to sites with the addition of 1 extra marked site, $M$, and using the fitted ON/OFF rates from the ODE model. The *Table 3* gives the reactions and propensities that are modelled.

Thirty runs over 60 min were used to generate mean and standard deviations with times to reach 95% of all sites at least once determined via fitting of an inverse exponential to the number of unique site visits counted via the variable $A_{CB}$. The time for available CLOCK:BMAL1 complexes to bind 95% of all binding sites at least once is calculated by fitting the recovery curve $f(t) = 1 - \exp(-\lambda t)$ to normalised stochastic trajectories of $S_{tot} - S$ ($S_{tot} = 3436$), see *Figure 7—figure supplement 1B*, and then converting the recovery rate $\lambda$ using the equation

$$\tau_{95\%} = \frac{\ln(20)}{\lambda}. \tag{17}$$

Visits per minute to a single site were calculated by counting binding and unbinding to $M$, which possesses the same ON and OFF rates as other target-sites. Assessment of the contribution of PER2 mediated displacement was performed by setting PER2 concentration to zero (KO) in the ODE model and using the simulated OFF rate in a parallel run to wild-type (WT) runs (*Figure 5—figure supplement 1B*), with the reduced number of visits attributed to the slower OFF rate. Furthermore, we observed the same behaviour in this reduced stochastic model, when compared to the full ODE model, for PER2 KO as the mean and standard deviation of the number of sites bound by CLOCK:BMAL1 in both WT and KO conditions, *Figure 7—figure supplement 1A*, being comparable to the ODE model results in (*Figure 5—figure supplement 1E*). Finally, to assess the differences that would be induced by different nuclear volumes, as seen between different cell types, we ran the stochastic model at the same molecular concentrations over two volumes; a small volume of 240 fl representative of a typical mouse embryonic fibroblast (MEF) or various immune cell types (see *Figure 3—figure supplement 1E*) and 926 fl as measured for our lung fibroblasts used throughout this study, *Figure 7—figure supplement 1C*. We note little difference in the rate at which CLOCK:BMAL1 visits the single marked site $M$, indicating that the increase in DNA sites comparatively to the number of molecules at a smaller nuclear volume was balanced by the increase in ON rate due to the now higher concentration of DNA.

## Acknowledgements

Supported by the Biotechnology and Biological Sciences Research Council, UK (awards BB/P017347/1 and BB/P017355/1 to ASIL and MHH), MRC core funding (MC_U105170643 to MHH), the National Institutes of Health (award GM107069 and GM141849 to CLP) and a University of California Office of the President Chancellor's Postdoctoral Fellowship (to JLF). ASIL is a Wellcome Investigator (107851/Z/15/Z) and AAK is supported by a Wellcome 4 year PhD studentship (216416/Z/19/Z).

## Additional information

### Funding

| Funder | Grant reference number | Author |
| --- | --- | --- |
| Biotechnology and Biological Sciences Research Council | BB/P017347/1 | James S Bagnall Nicola Begley Andrew SI Loudon |
| Biotechnology and Biological Sciences Research Council | BB/P017355/1 | Nicola J Smyllie Michael H Hastings |
| Medical Research Council | MC_U105170643 | Michael H Hastings |
| National Institutes of Health | GM107069 | Carrie L Partch |
| National Institutes of Health | GM141849 | Carrie L Partch |
| Wellcome Trust | 107851/Z/15/Z | Andrew SI Loudon |

| Funder | Grant reference number | Author |
|---|---|---|
| Wellcome Trust | 216416/Z/19/Z | Alex A Koch |
| University of California | | Jennifer L Fribourgh |

The funders had no role in study design, data collection and interpretation, or the decision to submit the work for publication.

## Author contributions

Alex A Koch, Conceptualization, Data curation, Formal analysis, Investigation, Methodology, Software, Visualization, Writing – original draft, Writing – review and editing; James S Bagnall, Conceptualization, Data curation, Formal analysis, Investigation, Methodology, Visualization, Writing – original draft, Writing – review and editing; Nicola J Smyllie, Formal analysis, Investigation, Methodology, Visualization, Writing – review and editing; Nicola Begley, Investigation; Antony D Adamson, Investigation, Resources, Validation; Jennifer L Fribourgh, Formal analysis, Investigation, Visualization; David G Spiller, Investigation, Methodology, Resources, Writing – review and editing; Qing-Jun Meng, Resources, Writing – review and editing; Carrie L Partch, Conceptualization, Formal analysis, Investigation, Resources, Visualization, Writing – review and editing; Korbinian Strimmer, Methodology, Software, Supervision; Thomas A House, Methodology, Software, Supervision, Validation, Writing – review and editing; Michael H Hastings, Conceptualization, Formal analysis, Funding acquisition, Investigation, Methodology, Project administration, Resources, Supervision, Validation, Writing – original draft, Writing – review and editing; Andrew SI Loudon, Conceptualization, Formal analysis, Funding acquisition, Methodology, Project administration, Supervision, Validation, Writing – original draft, Writing – review and editing

## Author ORCIDs

Alex A Koch ⓘ http://orcid.org/0000-0002-0592-331X
James S Bagnall ⓘ http://orcid.org/0000-0003-1735-5855
Nicola J Smyllie ⓘ http://orcid.org/0000-0002-6324-1421
Nicola Begley ⓘ http://orcid.org/0000-0003-3601-9152
Antony D Adamson ⓘ http://orcid.org/0000-0002-5408-0013
David G Spiller ⓘ http://orcid.org/0000-0003-2502-6787
Qing-Jun Meng ⓘ http://orcid.org/0000-0002-9426-8336
Carrie L Partch ⓘ http://orcid.org/0000-0002-4677-2861
Korbinian Strimmer ⓘ http://orcid.org/0000-0001-7917-2056
Thomas A House ⓘ http://orcid.org/0000-0001-5835-8062
Michael H Hastings ⓘ http://orcid.org/0000-0001-8576-6651
Andrew SI Loudon ⓘ http://orcid.org/0000-0003-3648-445X

## Ethics

All experimental procedures were carried out in accordance with the Animals (Scientific Procedures) Act of 1986, UK (Licence number PP7901495).

## Decision letter and Author response

Decision letter https://doi.org/10.7554/eLife.73976.sa1
Author response https://doi.org/10.7554/eLife.73976.sa2

---

# Additional files

## Supplementary files

• Transparent reporting form

## Data availability

Modelling and analtyical code has been made publicly available via GitHub. The FCS analysis software is at https://github.com/LoudonLab/FcsAnalysisPipeline (copy archived at swh:1:rev:b12e9007ed-7f8a033485e57c8605e27c67df74f1), and the modeling link is https://github.com/LoudonLab/CLOCK-BMAL1-DNA-Binding. Source Data files have been provided for all FCS measurements and FRAP measurements in Figures 1, 2, 3 ,4, and 6.

The following datasets were generated:

| Author(s) | Year | Dataset title | Dataset URL | Database and Identifier |
|-----------|------|---------------|-------------|-------------------------|
| Koch AA, Bagnall JS | 2021 | FCS Analysis Pipeline | https://github.com/LoudonLab/FcsAnalysisPipeline | GitHub, FcsAnalysisPipeline |
| Koch AA | 2021 | Modelling for Quantification of protein abundance and interaction defines a mechanism for operation of the circadian clock | https://github.com/LoudonLab/CLOCK-BMAL1-DNA-Binding | GitHub, CLOCK-BMAL1-DNA-Binding |

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
