## [Editor Report]

The transcriptional negative feedback loop of the mammalian circadian clock is mainly regulated by interactions among BMAL1, CLOCK, PER1/2 and CRY1/2 in the nucleus. While the binding of CRY with BMAL1:CLOCK is known to block the transcriptional activity of BMAL1:CLOCK and the binding of PER:CRY dissociates BMAL1:CLOCK from DNA have been known, our understanding is limited in qualitative level. Koch et al., quantified the dynamic interactions among the core clock molecules such as their diffusion coefficients, binding affinity, and abundances in the nucleus. This greatly improves our understanding of the mammalian circadian clock. Importantly, this dynamic information is incorporated via a mathematical model to understand BMAL1-CLOCK binding to the target site (e.g., circadian proteins operate within an optimal range to modulate E-box binding), providing a coherent view on the mechanism driving the oscillation.

---

## [Decision Letter]

**Decision letter after peer review:**

Thank you for submitting your article "Quantification of protein abundance and interaction defines a mechanism for operation of the circadian clock" for consideration by *eLife*. Your article has been reviewed by 2 peer reviewers, and the evaluation has been overseen by Naama Barkai as the Senior and Reviewing Editor. The reviewers have opted to remain anonymous.

The reviewers appreciate the study and find it suitable for publication in *eLife*, once the suggestions for revision below are followed. Please consider and address all specific comments of the reviewers.

*Reviewer #1 (Recommendations for the authors):*

Below are my specific questions and suggestions.

1) The authors report BMAL1 and CLOCK exhibit a 2:1 concentration ratio in the nucleus (Lines 127 and 353). However, only 10% percent of BMAL1 is part of the CLOCK/BMAL1 heterodimer. If only 10% BMAL1 with CLOCK and CLOCK/BMAL1 is heterodimer (Line 354), the BMAL1/CLOCK ratio should be about 9:1 to 5:1. If BMAL1 and CLOCK exhibit a 2:1 ratio in the nucleus, 50% percent of BMAL1 should be in the CLOCK/BMAL1 heterodimer. Is there an error in calculation?

2) This manuscript has total 8 figures. I think removing figure 3 to supplement figures will make paper much simpler and clear. Because figure 3 just talks about the Triple endogenous labelled mice exhibit normal circadian behavior and it is related to figure 4.

3) Lines 191-197 bout Figure 4, author said: "Interestingly, the mean interaction strength between BMAL1 and CRY1 is >2-fold stronger than that between BMAL1 and CLOCK (Figure 2G). A similar relationship was found in vitro when measuring interactions using biolayer interferometry (Fribourgh et al., 2020). This is therefore consistent with a model in which the low abundance of the CRY1 repressor is offset by a high affinity for the CLOCK:BMAL1." I fail to see the logical connection between the two observations. The author cite the "Fribourgh et al., 2020" paper which does not compare the interaction between CRY1/BMAL1 and CLOCK/BMAL1. That paper (1) compared the interaction between CRY1 and CRY2 with CLOCK/BMAL1; and PER2-CRY1 with CLOCK/BMAL1. A previous paper showed that CRY1 binding to CLOCK/BMAL1-E-box is dependent on CLOCK (2), not BMAL1 only. It was shown that CLOCK PAS-B docks into secondary pocket of CRY1(3), and it is known that CRY1 competes for binding with coactivators to the intrinsically unstructured C-terminal transactivation domain (TAD) of BMAL1 to establish a functional switch between activation and repression of CLOCK-BMAL1 (4). This means that CRY1 interacting with BMAL1 to repress must be with CLOCK on E-box, not BMAL1 only. However, here the interaction of CRY1 with BMAl1 does not represent CRY1 with CLOCK/BMAL1-E-box. Therefore, the observed interaction is most of CRY1 with free BMAL1 (10% percent of BMAL1 is part of the CLOCK/BMAL1 heterodimer, as authors said), which does not cause repression. The affinity BMAL1 to CRY1 (kD38.8) is much higher than PER2 to CRY1 (kD81.8) according to these kD vaules, which indicate that most of interaction of CRY1/BMAL1 is independent on CLOCK-E-box. Therefore, the statement that: "A rhythmic and strong interaction observed between BMAL1 and CRY1 facilitates repression" for figure 4 title is not justified.

4) Line 376, I recommend the authors cite the Ye et al., JBC, 2011 paper (2) instead of Akashi et al., 2014, which plagiarized the Ye et al., 2011 paper. The Ye et al., 2011 paper shows CRY1 binding to CLOCK/BMAL1-E-box ,PER2 removes CRY1 from CLOCK/BMAL1-E-box in vitro, and CRY inhibits CLOCK?BMAL1 even in the absence of PER in vivo for the first time.

5) Lines 57 to 58 and 200 to 209 about Figure 5: The authors' previous paper reported that: "PER2 …without circadian gating of nuclear localization" and "when co-expressed, subcellular localization was unchanged….(Figure 5A). PER2 mobility was not altered following co-expression with CRY1…(Figure 5B).". This is not consistent with previous work that shows that PER2 entry into nuclei is dependent on CRY1/2, because no PER2 can be observed in the nuclei in Cry1/2-/- mouse, but middle level of PER2 in the Cytoplasm (5,6). Can the authors comment on this?

1. Fribourgh, J. L., Srivastava, A., Sandate, C. R., Michael, A. K., Hsu, P. L., Rakers, C., Nguyen, L. T., Torgrimson, M. R., Parico, G. C. G., Tripathi, S., Zheng, N., Lander, G. C., Hirota, T., Tama, F., and Partch, C. L. (2020) Dynamics at the serine loop underlie differential affinity of cryptochromes for CLOCK:BMAL1 to control circadian timing. e*Life* 9

2. Ye, R., Selby, C. P., Ozturk, N., Annayev, Y., and Sancar, A. (2011) Biochemical analysis of the canonical model for the mammalian circadian clock. J Biol Chem 286, 25891-25902

3. Michael, A. K., Fribourgh, J. L., Chelliah, Y., Sandate, C. R., Hura, G. L., Schneidman-Duhovny, D., Tripathi, S. M., Takahashi, J. S., and Partch, C. L. (2017) Formation of a repressive complex in the mammalian circadian clock is mediated by the secondary pocket of CRY1. Proc Natl Acad Sci U S A 114, 1560-1565

4. Xu, H., Gustafson, C. L., Sammons, P. J., Khan, S. K., Parsley, N. C., Ramanathan, C., Lee, H. W., Liu, A. C., and Partch, C. L. (2015) Cryptochrome 1 regulates the circadian clock through dynamic interactions with the BMAL1 C terminus. Nat Struct Mol Biol 22, 476-484

5. Lee, C., Etchegaray, J. P., Cagampang, F. R., Loudon, A. S., and Reppert, S. M. (2001) Posttranslational mechanisms regulate the mammalian circadian clock. Cell 107, 855-867

6. Cao, X., Yang, Y., Selby, C. P., Liu, Z., and Sancar, A. (2021) Molecular mechanism of the repressive phase of the mammalian circadian clock. Proc Natl Acad Sci U S A 118

*Reviewer #2 (Recommendations for the authors):*

1. The authors utilized the following equation to calculate the average of the CLOCK:BMAL1-DNA binding OFF rate k_OFF and CLOCK:BMAL1CRY:PER2-DNA OFF rate R_OFF weighted by their respective relative concentration:

¯k_OFF=k_ON ([Unbound CB][Unbound Sites])/[Bound Sites] (1)

with

[Unbound CB]=[CB]_eq+[CBC1]_eq+[CBC1P2]_eq,

[Unbound Sites]=_eq,

[Bound sites]=[CBS]_eq+[CBC1S]_eq+[CBC1P2S]_eq.

Actually, the weighted average of the binding OFF rates is

k_OFF ([CBS]_eq+[CBC1S]_eq)/[Bound Sites] +R_OFF [CBC1P2S]_eq/[Bound Sites]

= k_ON ([CB]_eq _eq+[CBC1]_eq _eq)/[Bound Sites] +R_ON ([CBC1P2]_eq _eq)/[Bound Sites]

= [Unbound Sites]/[Bound Sites] {k_ON ([CB]_eq+[CBC1]_eq )+R_ON [CBC1P2]} (2).

Therefore, the ¯k_OFF is a correct equation for the weighted average of the binding OFF rates only when k_ON=R_ON. However, the estimated k_ON and R_ON in Table 1 are different. The authors had to use (2) instead of (1), or use (1) under the condition of k_ON=R_ON.

2. It is difficult to understand the process of parameter estimation of the ODE. Could you provide a table or figure to illustrate "the list of unknown parameters" and "the list of data used for the estimation"? How was the identifiability issue of the parameter estimation checked?

3. The authors explored the effect of the PER2 on the visiting rates of the CLOCK:BMAL1 to their binding sites using the reduced stochastic model. However, the stochastic model does not contain PER2. Using a full stochastic model including PER2 and all the interactions between proteins as in the ODE model can provide more reliable prediction.

4. PER1 and CRY2 were not considered in the model. How much is the model prediction affected by the missing?

---

## [Author Response]

Reviewer #1 (Recommendations for the authors):Below are my specific questions and suggestions.1) The authors report BMAL1 and CLOCK exhibit a 2:1 concentration ratio in the nucleus (Lines 127 and 353). However, only 10% percent of BMAL1 is part of the CLOCK/BMAL1 heterodimer. If only 10% BMAL1 with CLOCK and CLOCK/BMAL1 is heterodimer (Line 354), the BMAL1/CLOCK ratio should be about 9:1 to 5:1. If BMAL1 and CLOCK exhibit a 2:1 ratio in the nucleus, 50% percent of BMAL1 should be in the CLOCK/BMAL1 heterodimer. Is there an error in calculation?

The 2:1 ratio we used here describes the total nuclear concentration of BMAL1 to total nuclear CLOCK regardless of whether the protein is ‘free’ or in ‘complex’. It is correct to say that we find that only 10% of total BMAL1 is bound with CLOCK as a heterodimer. The calculations above by the referee may have assumed that the 2:1 ratio and the 10% are based on the same measure. We have no explanation for the 2:1 ratio here, but suggest that differences in protein turnover, import/export rates will likely contribute. To clarify this point, we have amended the text of the paper to read (lines 362-366):

“Our measures of total BMAL1 and CLOCK reveal a concentration ratio of 2:1, possibly reflecting differences in turnover rate, import and export of these two proteins. Strikingly, only 10% of BMAL1 was bound in complex with CLOCK. This ratio of 2:1 is compatible with recent modelling studies defining stoichiometric relationships within the nucleus (Lee et al., 2011; Kim and Forger, 2012)”

2) This manuscript has total 8 figures. I think removing figure 3 to supplement figures will make paper much simpler and clear. Because figure 3 just talks about the Triple endogenous labelled mice exhibit normal circadian behavior and it is related to figure 4.

We thank the referee for this suggestion and have amended the figures. We have retained a portion of Panel 3D from the original figure and incorporated this into the new main text figure as Panel 3A.

3) Lines 191-197 bout Figure 4, author said: "Interestingly, the mean interaction strength between BMAL1 and CRY1 is >2-fold stronger than that between BMAL1 and CLOCK (Figure 2G). A similar relationship was found in vitro when measuring interactions using biolayer interferometry (Fribourgh et al., 2020). This is therefore consistent with a model in which the low abundance of the CRY1 repressor is offset by a high affinity for the CLOCK:BMAL1." I fail to see the logical connection between the two observations. The author cite the "Fribourgh et al., 2020" paper which does not compare the interaction between CRY1/BMAL1 and CLOCK/BMAL1. That paper (1) compared the interaction between CRY1 and CRY2 with CLOCK/BMAL1; and PER2-CRY1 with CLOCK/BMAL1. A previous paper showed that CRY1 binding to CLOCK/BMAL1-E-box is dependent on CLOCK (2), not BMAL1 only. It was shown that CLOCK PAS-B docks into secondary pocket of CRY1(3), and it is known that CRY1 competes for binding with coactivators to the intrinsically unstructured C-terminal transactivation domain (TAD) of BMAL1 to establish a functional switch between activation and repression of CLOCK-BMAL1 (4). This means that CRY1 interacting with BMAL1 to repress must be with CLOCK on E-box, not BMAL1 only. However, here the interaction of CRY1 with BMAl1 does not represent CRY1 with CLOCK/BMAL1-E-box. Therefore, the observed interaction is most of CRY1 with free BMAL1 (10% percent of BMAL1 is part of the CLOCK/BMAL1 heterodimer, as authors said), which does not cause repression. The affinity BMAL1 to CRY1 (kD38.8) is much higher than PER2 to CRY1 (kD81.8) according to these kD vaules, which indicate that most of interaction of CRY1/BMAL1 is independent on CLOCK-E-box. Therefore, the statement that: "A rhythmic and strong interaction observed between BMAL1 and CRY1 facilitates repression" for figure 4 title is not justified.

The reviewer has highlighted an important distinction. Our FCCS measurement of interaction between BMAL1 and CRY1 does not specifically distinguish whether they are the sole elements of the ‘complex’ or whether additional ‘un-labelled’ proteins are also present, as would be the case for CLOCK. However, in this specific analysis our diffusion data of BMAL1 also shows a reduced mobility similar to that when co-expressed alongside CLOCK (as observed in Figure 2B) lending further support that we are likely seeing the interaction of CRY1 with CLOCK/BMAL1.

We have altered the figure title and addressed these issues in the text to provide clarity (lines 200-203):

“Although, these interaction measurements do not distinguish between whether either proteins are complexed with other partners, diffusion data are consistent with BMAL1 being bound to CLOCK, and are compatible with a model in which the low abundance of the CRY1 repressor is offset by a high affinity for the CLOCK:BMAL1 heterodimer”

Proposed figure title "A rhythmic and strong interaction observed between slow-diffusing BMAL1 and CRY1 facilitates repression"

4) Line 376, I recommend the authors cite the Ye et al., JBC, 2011 paper (2) instead of Akashi et al., 2014, which plagiarized the Ye et al., 2011 paper. The Ye et al., 2011 paper shows CRY1 binding to CLOCK/BMAL1-E-box ,PER2 removes CRY1 from CLOCK/BMAL1-E-box in vitro, and CRY inhibits CLOCK?BMAL1 even in the absence of PER in vivo for the first time.

Thank you for pointing this out and it is quite correct that we replace the Akashi et al., citation with the Ye et al., 2011 on the basis this work preceded Akashi and similarly supports our statements.

5) Lines 57 to 58 and 200 to 209 about Figure 5: The authors' previous paper reported that: "PER2 …without circadian gating of nuclear localization" and "when co-expressed, subcellular localization was unchanged….(Figure 5A). PER2 mobility was not altered following co-expression with CRY1…(Figure 5B).". This is not consistent with previous work that shows that PER2 entry into nuclei is dependent on CRY1/2, because no PER2 can be observed in the nuclei in Cry1/2-/- mouse, but middle level of PER2 in the Cytoplasm (5,6). Can the authors comment on this?

PER2 mobility, as measured by diffusion speed, was found to be insensitive to additional CRY1. It is worth noting that additional protein would typically slow the complex further and that PER2 already moves at a very slow speed ~0.2 μm/s so any further reduction would be very difficult to detect with our system. Additionally, we have recently collaborated on a paper (Smyllie et al., 2022 PNAS) that demonstrates that only a relatively small amounts of CRY1 are needed to localize PER2 to the nucleus. CRY1 never reaches a low enough nadir to disrupt PER2 localization, and this may explain why we see no gating on nuclear localization of PER2. We have clarified this in the introduction as below and added the recent the Smyllie et al., 2022 reference (lines 60-62):

“Only a relatively small amount of CRY1 is needed to localize PER2 to the nucleus, as shown in live SCN studies, with PER2 localization remaining predominantly nuclear throughout the day (Smyllie et al., 2022).”

Reviewer #2 (Recommendations for the authors):1. The authors utilized the following equation to calculate the average of the CLOCK:BMAL1-DNA binding OFF rate k_OFF and CLOCK:BMAL1CRY:PER2-DNA OFF rate R_OFF weighted by their respective relative concentration:¯k_OFF=k_ON ([Unbound CB][Unbound Sites])/[Bound Sites] (1)with[Unbound CB]=[CB]_eq+[CBC1]_eq+[CBC1P2]_eq,[Unbound Sites]=_eq,[Bound sites]=[CBS]_eq+[CBC1S]_eq+[CBC1P2S]_eq.Actually, the weighted average of the binding OFF rates isk_OFF ([CBS]_eq+[CBC1S]_eq)/[Bound Sites] +R_OFF [CBC1P2S]_eq/[Bound Sites]=k_ON ([CB]_eq _eq+[CBC1]_eq _eq)/[Bound Sites] +R_ON ([CBC1P2]_eq _eq)/[Bound Sites]=[Unbound Sites]/[Bound Sites] {k_ON ([CB]_eq+[CBC1]_eq )+R_ON [CBC1P2]} (2).Therefore, the ¯k_OFF is a correct equation for the weighted average of the binding OFF rates only when k_ON=R_ON. However, the estimated k_ON and R_ON in Table 1 are different. The authors had to use (2) instead of (1), or use (1) under the condition of k_ON=R_ON.

We thank the reviewer for picking up this discrepancy. We have since, set k_ON=R_ON in our model and this eliminated R_ON as an independent parameter to fit. This has made little difference to the fit and therefore the conclusions stemming from the modelling. The figures containing model derived values as well as the new table of fitted parameters have been updated to reflect this reparameterization.

2. It is difficult to understand the process of parameter estimation of the ODE. Could you provide a table or figure to illustrate "the list of unknown parameters" and "the list of data used for the estimation"?

Thank you for pointing out this issue. Table 1 has also been reformulated to aid understanding by splitting out the unknown parameters and subsequent fit values from the list of data used. We have also listed the derived parameters.

How was the identifiability issue of the parameter estimation checked?

The reviewer has pointed out the important issue of issue of identifiability. We have checked this by calculating the eigenvalues of the inverse Hessian matrix of the fit, finding that it is non-singular and reasonably well-conditioned. We have added the following text to lines (250 – 252):

“In order to confirm identifiability of the unknown parameters we calculated the eigenvalues of the Hessian matrix of the fit, finding that it is non-singular and reasonably well conditioned (Table 1).”

3. The authors explored the effect of the PER2 on the visiting rates of the CLOCK:BMAL1 to their binding sites using the reduced stochastic model. However, the stochastic model does not contain PER2. Using a full stochastic model including PER2 and all the interactions between proteins as in the ODE model can provide more reliable prediction.

We thank the referee for this comment and would like to clarify why we believe the reduced model better uses our data. FRAP data have allowed us to infer the kinetics of DNA binding reactions alongside FCCS measurements of protein-protein complexing. However, FCCS is unable to measure association and disassociation rates, only the ratio K_D_. In contrast to the ODE model, conclusions drawn from the full stochastic model are sensitive to these experimentally unavailable rates. As we are interested in DNA binding events involving CLOCK and BMAL1 (which we measure), a reduced model using measured changes in DNA binding yields more reliable predictions of CLOCK:BMAL1 binding kinetics based on our data.

4. PER1 and CRY2 were not considered in the model. How much is the model prediction affected by the missing?

We thank the referee for the option to clarify, we assume this is reference to the stochastic model and here the absence of PER1 and CRY2 does not impact. The absence of PER1 and CRY2 does not impact our stochastic modelling, which only is concerned with the effect of the measured BMAL1 DNA binding rhythms and therefore already encompasses the source without explicitly detailing how much is attributed to PER1 or CRY2.